# Shade-induced RTFL/DVL peptides negatively regulate the shade response by directly interacting with BSKs in Arabidopsis

Sha Huang[1], Yu Ma[1], Yitian Xu[1], Pengfei Lu[1], Jie Yang[2], Yu Xie[1], Jianhua Gan [ORCID][2] & Lin Li [ORCID][1]✉

For shade-intolerant species, shade light indicates the close proximity of neighboring plants and triggers the shade avoidance syndrome (SAS), which causes exaggerated growth and reduced crop yield. Here, we report that non-secreted ROT FOUR LIKE (RTFL)/DEVIL (DVL) peptides negatively regulate SAS by interacting with BRASSINOSTEROID SIGNALING KINASEs (BSKs) and reducing the protein level of PHYTOCHROME INTERACTING FACTOR 4 (PIF4) in Arabidopsis. The transcription of at least five *RTFLs* (*RTFL13/16/17/18/21*) is induced by low R:FR light. The RTFL18 (DVL1) protein is stabilized under low R:FR conditions and localized to the plasma membrane. A phenotype analysis reveals that RTFL18 negatively regulates low R:FR-promoted petiole elongation. BSK3 and BSK6 are identified as partners of RTFL18 through binding assays and structural modeling. The overexpression of *RTFL18* or knockdown of *BSK3/6* reduces BRASSINOSTEROID signaling and reduces low R:FR-stabilized PIF4 levels. Genetically, the overexpression of *BSK3/6* and *PIF4* restores the petiole phenotype acquired by *RTFL18*-overexpressing lines. Collectively, our work characterizes a signaling cascade (the RTFLs-BSK3/6-PIF4 pathway) that prevents the excessive activation of the shade avoidance response in Arabidopsis.

Plants are sessile organisms that cannot escape their environment. Therefore, it is crucial for plants to sense their surroundings and initiate appropriate responses to changing environments. Light provides energy for photosynthesis and also acts as an important environmental signal to affect growth and development throughout the whole life cycle of plants. Because chlorophyll mainly absorbs red and blue light during photosynthesis, shade-intolerant plants sense low red-to-far-red light ratios (low R:FR). As a result, crowded plants undergo several morphological and physiological changes via shade avoidance syndrome (SAS)[1]. A typical SAS involves increased hypocotyl and petiole elongation, a rapid arrest of leaf development, fewer instances of branching, early flowering, increased susceptibility to various diseases, and limited crop yield[2–5].

Phytochromes are the main photoreceptors that sense red and far-red light. In *phytochrome B (phyB)* mutant-carrying Arabidopsis, the constitutive SAS phenotype indicates that phyB plays a dominant role in inhibiting the SAS[6]. The PHYTOCHROME-INTERACTING FACTORs (PIFs) belong to a subfamily of basic helix-loop-helix (bHLH) transcription factors. PIF4, PIF5, and PIF7 have been implicated downstream of phyB. Shade light promotes the dephosphorylation of PIF7 and the accumulation of PIF4/PIF5 proteins, which activate the transcription of downstream genes that mediate shade responses[7–10].

Moreover, various phytohormones are involved and function in a coordinated manner to shape shade-regulated plant architecture[11]. For instance, BRASSINOSTEROID (BR)-biosynthetic dwarf mutants (*dwarf1, sav1,* and *rot3*) display defects in shade-induced hypocotyl

[1]State Key Laboratory of Genetic Engineering, Institute of Plants Biology, School of Life Sciences, Fudan University, Shanghai 200438, China. [2]Shanghai Public Health Clinical Center, State Key Laboratory of Genetic Engineering, Department of Physiology and Biophysics, School of Life Sciences, Fudan University, Shanghai 200438, China. ✉e-mail: linli@fudan.edu.cn

elongation[12–14]. The BR signaling components, BR-ENHANCED EXPRESSION (BEE) and BES1-INTERACTING MYC-LIKE (BIM), are also involved in the regulation of the SAS[15]. More importantly, PIF4 is a phosphorylation target of BRASSINOSTEROID-INSENSITIVE 2 (BIN2), which marks PIF4 for proteasome degradation[16]. Recently, shade light has been reported to enhance BR signaling through *BSK5* and lead to the activation of BRI1-EMS-SUPPRESSOR 1 (BES1)[17]. Intact BR biosynthesis and the BR signaling pathway are required for SAS activity.

Peptides are important signaling molecules in intercellular and intracellular communications for plant growth and environmental responses[18]. Based on peptide structure, a bioinformatics analysis has revealed more than 7000 novel candidate small coding genes in Arabidopsis, and some of these genes are likely to be associated with hormone-like peptides[19]. Although a number of peptides have been shown to participate in plant growth, development[20–23] and environmental responses[24,25], to date, only very few of these peptides have been functionally characterized or matched to a receptor.

*ROTUNDIFOLIA4* (*ROT4*), encoding a 53 amino acid peptide, was isolated through gain-of-function genetic screening for proteins associated with "small-round" rosette leaves[26]. In the same year, *DEVIL1* (*DVL1*), encoding a 51 amino acid polypeptide, was identified via gain-of-function genetic screening for genes that influence fruit development[27] and is a paralog of *ROT4*. Overexpression of *DVL1* results in the acquisition of pleiotropic phenotypes, including shortened stature, rounder rosette leaves, clustered inflorescences, shortened pedicles, and fruits with pronged tips. In Arabidopsis, more than 20 putative homologs of the *ROT4* and *DVL1* genes have been identified, and they constitute the *ROT FOUR LIKE/DEVIL* (*RTFL/DVL*) family[28]. Ectopic overexpression of each of these five closely related *RTFL/DVL* genes leads to similar phenotypic changes[27]. RTFL/DVL proteins carry a conserved 30-amino-acid region located in the C-terminus and named the RTF domain, which seems to be sufficient to induce the acquisition of this phenotype[26]. RTFL/DVL peptides do not carry a signal peptide, and their N-terminal regions are not highly conserved or studied. To date, the mechanism of action of RTFL/DVL peptides remains unknown. The overexpression of *RTFL/DVL* genes results in downregulation of *FRUITFUL* (*FUL/AGL8*), a MADS-box gene involved in valve differentiation[27], and the loss of function of *NAC1* reverses the *DVL* overexpression-related phenotype to a WT-like phenotype[29].

In the current study, we find that low R:FR light induce the transcription of at least five *RTFL/DVL* peptides. RTFL18 (DVL1) is specifically expressed in leaf blades and petioles and localized to the plasma membrane. RTFL18 is involved in low R:FR-promoted petiole elongation, which disrupted the function of BSK3/6 through direct interaction. Inhibited BR signaling and reduced PIF4 protein levels are found in *RTFL18ox* and *bsk3/6* mutants. Overexpression of *PIF4* can rescue the petiole phenotype of *RTFL18*-overexpressing lines. These findings identify the molecular mechanism of non-secreted RTFL/DVL peptides in the shade avoidance response, establish a link between shade light and the BR signaling pathway, and provide putative targets for engineering density tolerant crops.

## Results

### Low R:FR light significantly regulates the transcription of at least five *RTFL/DVL* genes

Arabidopsis RTFL/DVL peptides constitute a family consisting of 24 members (Supplementary Fig. 1a). *RTFL13, RTFL15, RTFL16, RTFL17, RTFL18,* and *RTFL21* were considered shade-induced genes based on published RNA-sequencing dataset (Fig. 1a). We performed RT–qPCR to confirm the low R:FR induction of *RTFL13, RTFL16, RTFL17, RTFL18,* and *RTFL21* (Fig. 1b, c, and Supplementary Fig. 1b, c). We also found that their transcripts are repressed from darkness to white light (Fig. 1d) and induced in *phyB-9^{BC}* (Fig. 1e), indicating the transcripts of these *RTFLs* were negatively regulated by light and phyB.

The low R:FR-induced expression of these five *RTFL* genes was decreased in *pif7-1, pif7-2,* and *pifq* (*pif1pif3pif4pif5*) mutant plants, suggesting that their transcripts were positively regulated by PIFs (Fig. 1f). By evaluating published ChIP-sequencing data on PIF7 and PIF4[30,31], we found that PIF7 and PIF4 were enriched in the G-box motif of the *RTFL16, RTFL17, RTFL18,* and *RTFL21* gene promoters (Fig. 1g). We verified binding of PIF7 and PIF4 to the promoters of *RTFL18* and *RTFL21* through a ChIP–qPCR assay (Fig. 1h).

These data reveal that at least five *RTFL/DVL* transcripts are increased under low R:FR conditions, suggesting their possible regulatory roles in the SAS.

### Expression profile and subcellular localization of RTFL18

Among the five *RTFL/DVL* genes, *RTFL18* was maximally upregulated upon low R:FR exposure. Therefore, we focused on the characteristics of RTFL18 in the following work. In transgenic lines that express luciferase driven by the *RTFL18* promoter (*pRTFL18::Luc*), *RTFL18* was mainly expressed in leaf blades and petioles, but was also expressed in hypocotyls, roots, flowers, or fruits (Fig. 2a). RT–qPCR data confirmed the low R:FR-inductions of *RTFL18* in hypocotyls, cotyledons & petioles, and leaf blades & petioles (Fig. 2b). The expression pattern of *RTFL18* was consistent with that reported in publicly available reports (Supplementary Fig. 2a). Low R:FR induced the expressions of *RTFL13/16/21* in hypocotyls, cotyledons & petioles, and leaf blades & petioles, while *RTFL17* was mainly induced in cotyledons & petioles (Supplementary Fig. 2b).

To determine the protein stability of RTFL18, we generated *35 S::Flag-RTFL18* transgenic lines. The Flag-RTFL18 protein in three independent lines accumulated under low R:FR conditions (30, 90, and 180 min treatments) (Fig. 2c).

To determine its subcellular localization, GFP-RTFL18 was transiently expressed in *Nicotiana benthamiana* leaves. As shown in Fig. 2d, GFP-RTFL18 was localized to the plasma membrane and, in merged images, clearly colocalized with stained FM4−64, an endocytic tracer and plasma membrane dye. Immunofluorescence assays indicated that Flag-RTFL18 in the *35 S::Flag-RTFL18* transgenic line was localized on the plasma membrane (Fig. 2e and Supplementary Fig. 2c). Using a Minute™ Plasma Membrane Protein Isolation Kit for Plant to separate various components in cells, the presence of Flag-RTFL18 from the *35 S::Flag-RTFL18* transgenic line was detected only in the membrane (Fig. 2f). These results indicate that RTFL18 is localized on the plasma membrane.

### RTFL18 negatively regulates low R:FR-promoted petiole elongation

To explore the functions of RTFL18, we obtained an activation-tagged line *dvl1-1D*, which contains a T-DNA insertion into the promoter of *RTFL18*, establishing an overexpression allele[27]. Consistent with the findings from experiments with *dvl1-1D* and *35 S::RTFL18* transgenic lines, in experiments with *35 S::Flag-RTFL18*, *35 S::RTFL18-Flag*, *35 S::RTFL21* and *35 S::RTFL21-Flag* overexpression, the results showed shorter cotyledon petiole and first and second leaf petiole lengths under low R:FR conditions (Fig. 3a–c and Supplementary Fig. 3a, b). Although shorter hypocotyl lengths were also observed in these *RTFLs*-overexpressing lines (Supplementary Fig. 3c−e), we primarily focused on petiole elongation due to the higher tissue-specific expression of *RTFL18*.

We also generated *rtfl18, rtfl21* single mutants and *rtfl18*rtfl21* double mutants by the CRISPR–Cas9 gene editing system. One-nucleotide insertion and 10-nucleotide deletion on the exon of *RTFL18* were found in *rtfl18-1, rtfl18-2,* and *rtfl18-3*. One-nucleotide insertion of *RTFL18* and *RTFL21* was found in *rtfl18*rtfl21*, and one-nucleotide insertion of *RTFL21* was found in *rtfl21*. These mutations generated premature stop codons and thus led to truncated proteins due to a coding frame shift (Fig. 3d). Only the double mutant showed a

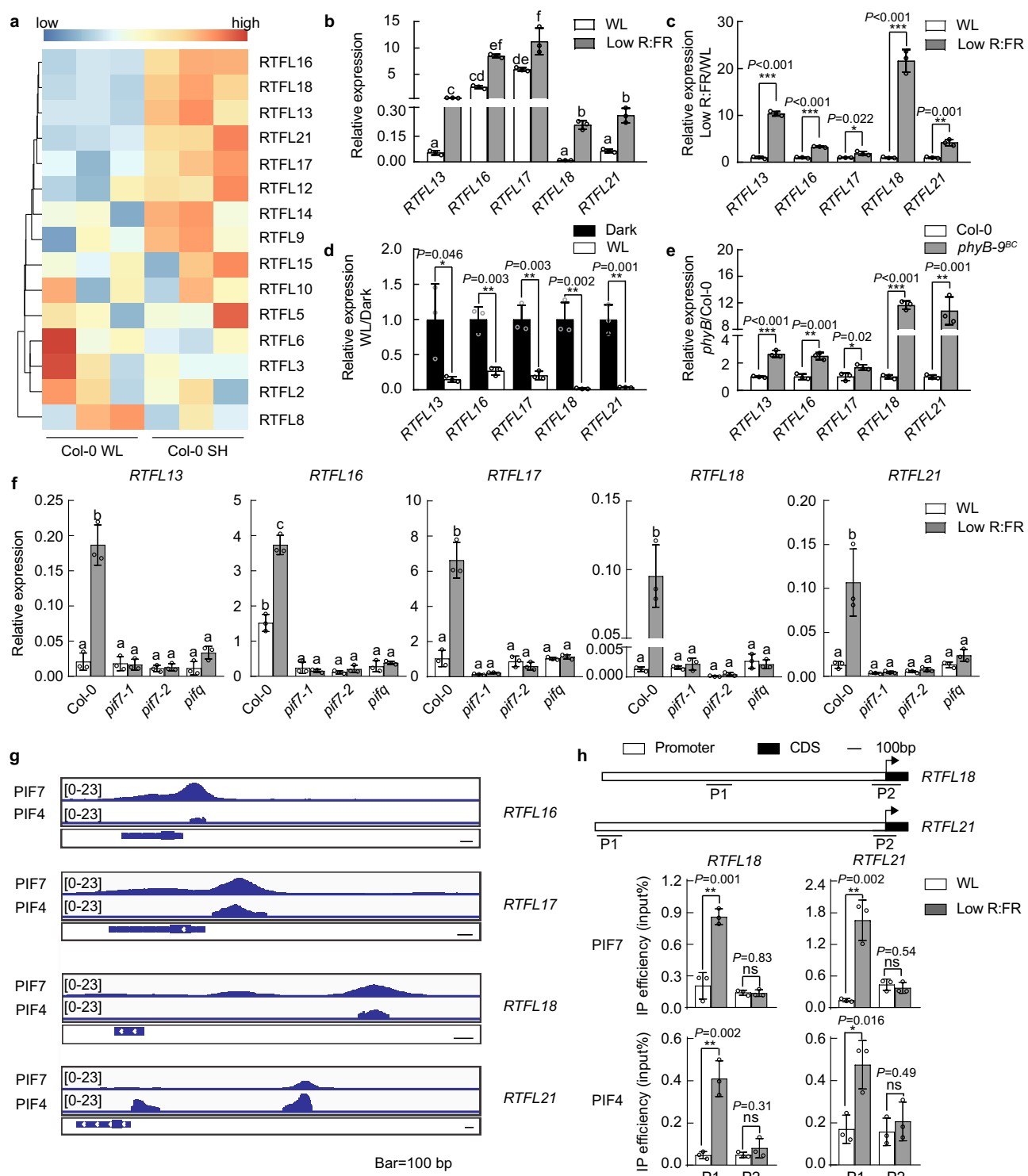

**Fig. 1 | Low R:FR light significantly regulates the transcription of at least five *RTFL/DVL* genes. a** Heatmap showing the transcript levels of *RTFLs* under white light (WL) and shade light (SH) based on GSE146125. Red, yellow and blue rows indicate RNA expression at high, medium and low levels, respectively. **b** The relative expression levels of *RTFLs* under WL and low R:FR conditions. **c** Low R:FR induction of *RTFLs*. The expression levels of *RTFLs* under WL were standardized to be "1". **d** The relative expression levels of *RTFLs* under dark and WL conditions. The expression levels of the *RTFLs* in the dark were standardized to be "1". **e** The relative expression levels of *RTFLs* in Col-0 and *phyB-9^{BC}* seedlings. The expression levels of *RTFLs* in Col-0 were standardized as "1". **f** The relative expression levels of *RTFLs* in Col-0, *pif7-1*, *pif7-2*, and *pifq* seedlings. **g** Integrative Genomics Viewer (IGV) screenshots showing the enrichment of PIF7 and PIF4 on the promoters of *RTFL18* and *RTFL21* based on the GSE156584 and GSE35315 datasets. The scale bar indicates

100 bp. **h** ChIP–qPCR analyses confirming the enrichment of PIF7 and PIF4 on the *RTFL18* and *RTFL21* promoters in *PIF7ox* (*35 S::PIF7-Flash*(9 × Myc-6 × His-3 × Flag)*) and *PIF4ox*(*35 S::PIF4-Flash*(9 × Myc-6 × His-3 × Flag)*) seedlings. Data are presented as mean values +/− SD (*n* = 3, *n* refers to biological replicates). In **b**–**f**, the expression levels of *RTFLs* were normalized against the expression of the reference gene *AT2G39960*. Data are presented as mean values +/− SD (*n* = 3, *n* refers to biological replicates). The asterisks indicate significant differences to WL (**c**, **h**), dark (**d**) and Col-0 (**e**), respectively (Multiple *t* test: False Discovery Rate approach, *\*P* < 0.05, \*\**P* < 0.01, \*\*\**P* < 0.001, ns indicates no significance). Letters indicate significant differences between mean values (**b** one-way ANOVA, *P* < 0.01; **f** two-way ANOVA, *P* < 0.01), and groups with the same letters are not significantly different. Source data are provided as a Source Data file.

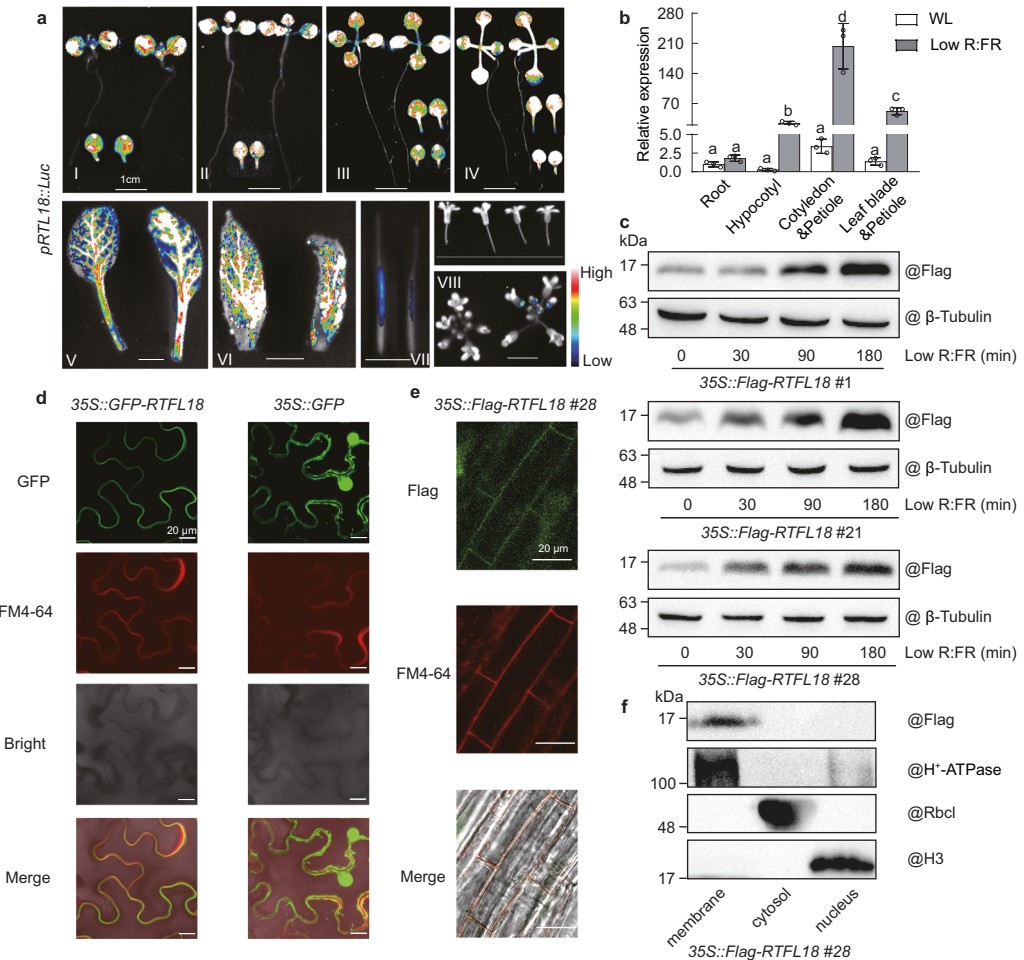

**Fig. 2 | Expression profile and subcellular localization of RTFL18. a** The expression of *RTFL18* in different tissues was determined on the basis of luciferase signal intensity in *pRTFL18::Luc* transgenic lines. I, III, seedlings were grown under white light for 8 (I) or 11 (III) days; II, IV seedlings were grown under white light for 3 days and transferred to low R:FR and grown for 5 (II) or 8 (IV) days; V, rosette leaves from plants grown under long day conditions for 4 weeks; VI, cauline leaves, VII, fruits, and VIII, flowers from plants grown under long day conditions for 6 weeks. **b** The relative expression of *RTFL18* in different tissues under WL and low R:FR conditions was measured by RT–qPCR. The expression in the roots under WL was standardized to be "1". Data are presented as mean values +/− SD ($n$ = 3, $n$ refers to biological replicates). Letters indicate significant differences between mean values (one-way ANOVA, $P < 0.01$), and groups with the same letters are not significantly different. **c** The protein levels of RTFL18 under WL and low R:FR conditions were measured by western blotting. Whole seedlings were used for experiments. Western blots were probed with anti-Flag antibody. The levels of β-tubulin indicated the loading control. **d** Subcellular localization of GFP-RTFL18 was measured by transient expression assay in *Nicotiana benthamiana* leaves. **e** Subcellular localization of Flag-RTFL18 was measured by immunolocalization assay in Arabidopsis seedlings. Flag-RTFL18 (green) was labeled by immunolocalization using anti-Flag antibodies and observed with a confocal laser scanning microscope. The membrane was stained with FM4−64 (red). GFP, GFP signal; FM4−64, FM4−64 staining; Flag, Flag signal; Bright, bright field; Merge, merged image of the GFP/Flag signal with the FM4−64 signal in a bright field. **f** Subcellular localization of Flag-RTFL18 was measured by western blotting. Western blots were probed with anti-Flag, anti-H⁺ −ATPase (Plasma membrane marker), anti-Rbcl, and anti-H3 antibodies. Scale bars represent 1 cm (**a**), and 20 μm (**d**, **e**). In **c**–**f**, each experiment was repeated three times with similar results. Source data are provided as a Source Data file.

longer petiole length than the wild-type plants under low R:FR condition (Fig. 3e–g), and there was no significant difference in hypocotyl elongation (Supplementary Fig. 3f). We further found that the changes in petiole elongation were reflected in epidermal cell elongation in *dvl1-1D* and *rtfl18*rtfl21* (Supplementary Fig. 3g).

These data suggest that RTFL18 negatively regulates low R:FR-promoted petiole elongation.

## RTFL18 interacts with BSK3 and BSK6

To elucidate the molecular mechanism by which RTFLs are involved in SAS, we screened RTFL18 interacting partners by immunoprecipitation-mass spectrometry (IP-MS) using Flag-tagged *RTFL18* transgenic lines (Supplementary Fig. 4a). BRASSINOSTEROID SIGNALING KINASE (BSK) family members appeared in the candidate lists (Supplementary Fig. 4b). BSKs belong to the RLCK-VII subfamily and are partially redundant positive regulators of BR signaling in Arabidopsis[32]. Considering the plasma membrane localization of BSKs, we first confirmed the interactions between RTFLs and BSK3/6. In a tobacco LCI assay, BSK3 or BSK6 fused with the N-terminal half of luciferase (Luc) interacted with RTFL18 or RTFL21 fused with the C-terminal half of Luc, generating strong Luc fluorescence, whereas no Luc activity signal was detected in the negative controls (Fig. 4a and Supplementary Fig. 4c). In a BiFC assay, BSK3 or BSK6 fused with the C-terminal half of yellow fluorescent protein (YFP) interacted with RTFL18 fused with the N-terminal half of YFP, generating a YFP signal in the cell membrane, whereas no YFP signal was detected in the negative controls (Fig. 4b). In a semi-in vivo pull-down assay, total proteins extracted from *N. benthamiana* leaves expressing BSK3-Flag and BSK6-Flag fusion proteins could be pulled down by GST-RTFL18 protein but not by the GST tag alone (a negative control) (Fig. 4c, d). In an in vitro pull-down assay, GST-BSK3 and GST-BSK6 fusion proteins were pulled down by Sumo-His-RTFL18, but GST could not (Fig. 4e). Moreover,

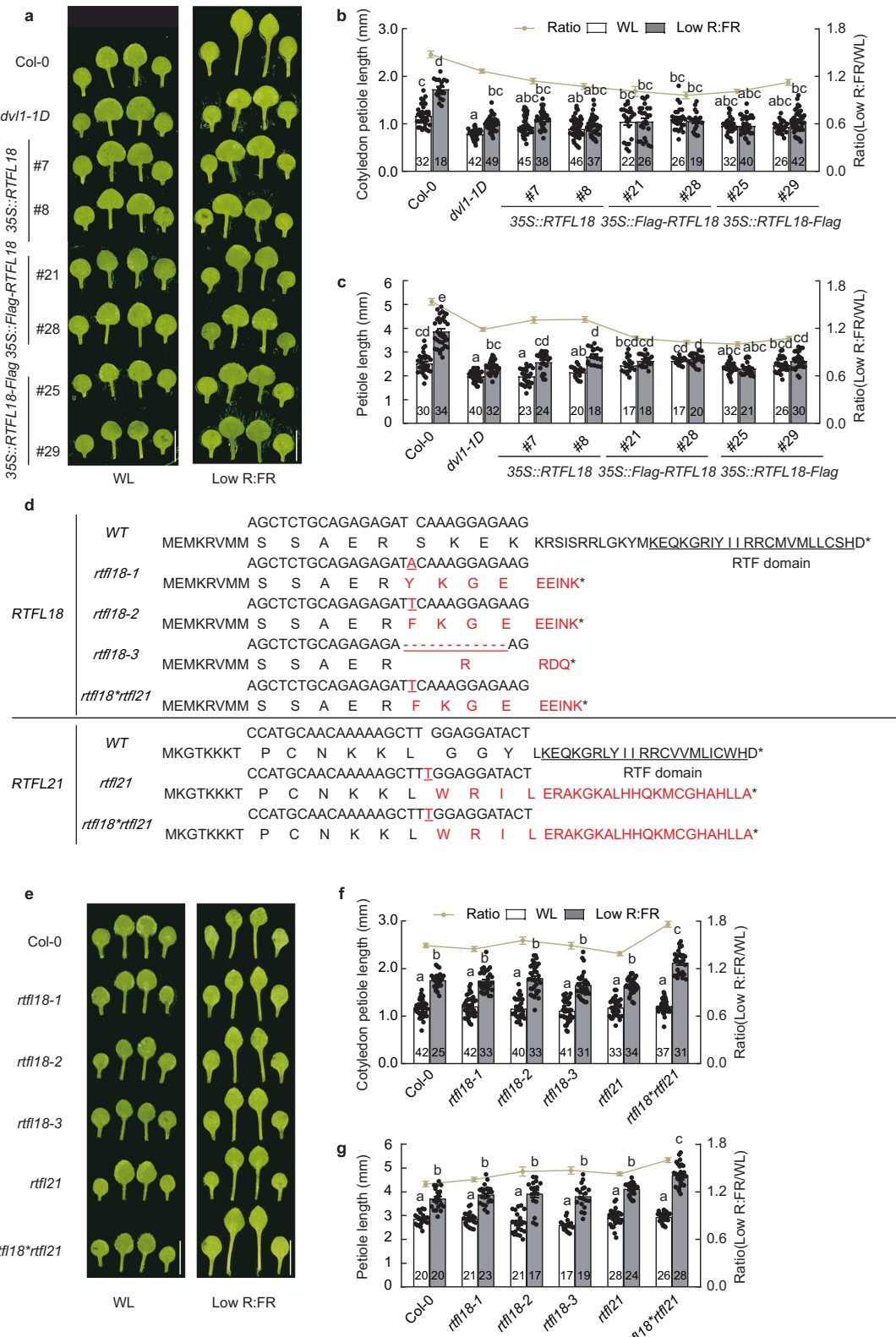

**Fig. 3 | RTFL18 negatively regulates low R:FR-promoted petiole elongation.**
**a** Pictures of Col-0, *dvl1-1D*, *35S::RTFL18* (#7, #8), *35S::Flag-RTFL18* (#21, #28), and *35S::RTFL18-Flag* (#25, #29) seedlings grown under white light (WL) and low R:FR conditions. **b** The petiole lengths of the cotyledons in Col-0, *dvl1-1D*, *35S::RTFL18*, *35S::Flag-RTFL18*, and *35S::RTFL18-Flag*. **c** The petiole lengths of the first and second leaves in the Col-0, *dvl1-1D*, *35S::RTFL18*, *35S::Flag-RTFL18*, and *35S::RTFL18-Flag* plants grown under WL and low R:FR conditions. **d** The mutants of *rtfl18*, *rtfl21*, and *rtfl18*rtfl21* were generated by the CRISPR/Cas9 system. The red parts indicate inserted or missing bases. The black underlined sections represent the RTF domain.

**e** Pictures of Col-0, *rtfl18*, *rtfl21*, and *rtfl18*rtfl21* lines grown under WL and low R:FR conditions. **f** The petiole lengths of the cotyledons in the Col-0, *rtfl18*, *rtfl21*, and *rtfl18*rtfl21* lines under WL and low R:FR conditions. **g** The petiole lengths of the first and second leaves in the Col-0, *rtfl18*, *rtfl21*, and *rtfl18*rtfl21* plants grown under WL and low R:FR conditions. The numbers in the bar charts represent *n* of each sample. Data are presented as mean values +/− SEM. Letters indicate significant differences between mean values (two-way ANOVA: Tukey's multiple comparisons test, $P < 0.01$), and groups with the same letters are not significantly different. Scale bars represent 5 mm. Source data are provided as a Source Data file.

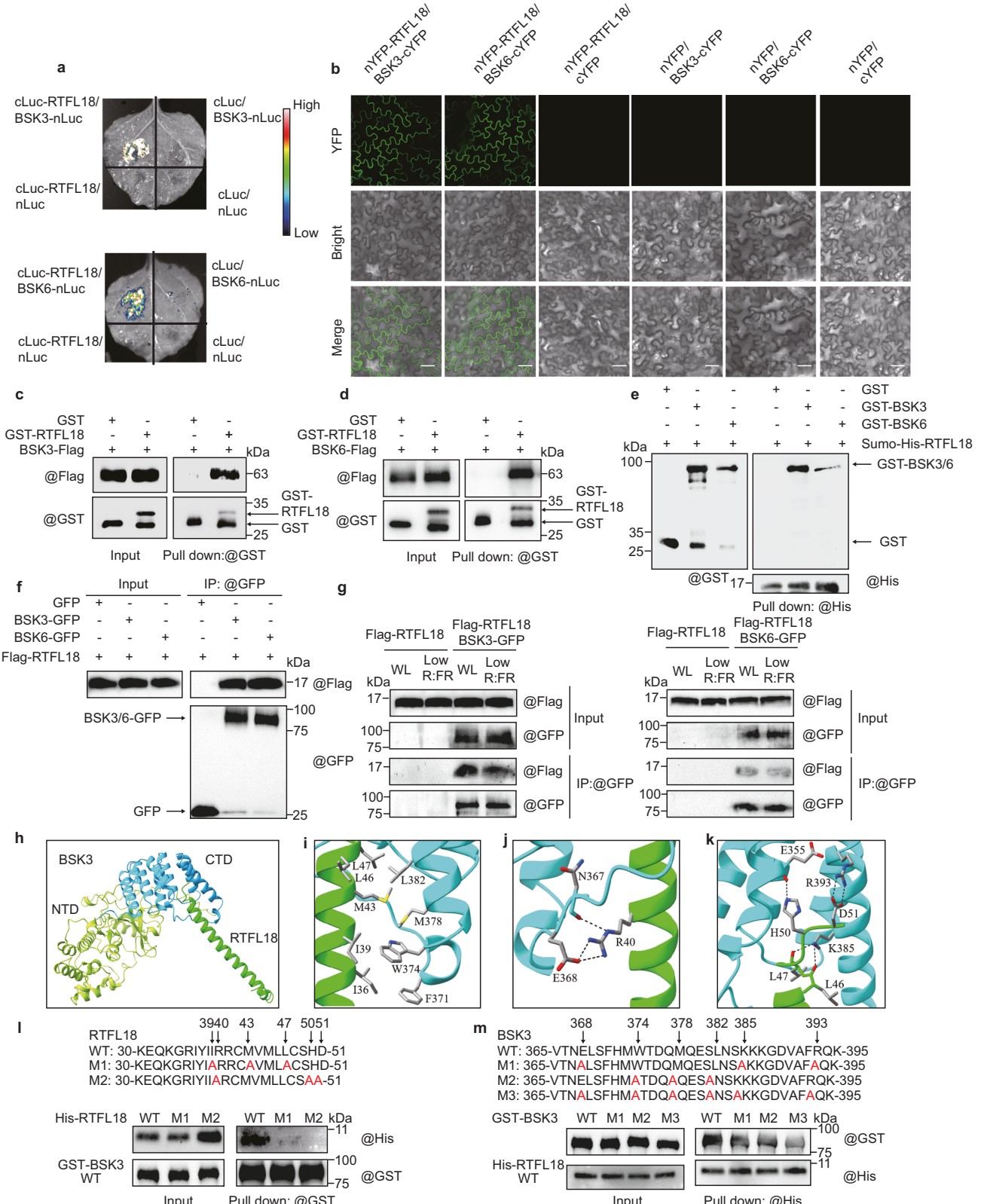

Flag-RTFL18 proteins were immunoprecipitated with BSK3-GFP or BSK6-GFP, but not GFP only in *N. benthamiana* leaves (Fig. 4f). The co-immunoprecipitation between BSK3/6-GFP and Flag-RTFL18 also can be detected in Arabidopsis *Flag-RTFL18*BSK3-GFP* and *Flag-RTFL18*BSK6-GFP* double overexpressing lines (Fig. 4g).

To gain further information about the interaction, we analyzed the structure of the RTFL18-BSK3/6 complex by AlphaFold-Multimer.

As suggested by the model (Fig. 4h), BSK3 can be divided into two domains: the N-terminal domain (NTD) and the C-terminal domain (CTD). RTFL18 forms one long helix, and its C-terminal residues can form various types of interactions with the BSK3 CTD, including hydrophobic interactions between the side chains of I36, I39, M43, L46, and L47 of RTFL18 and F371, W374, M378, and L382 of BSK3 (Fig. 4i) and hydrogen bond (H-bond) interactions between R40 of

**Fig. 4 | RTFL18 interacts with BSK3 and BSK6. a** Interactions between RTFL18 and BSK3/BSK6 were detected by luciferase complementary imaging (LCI). **b** Interactions between RTFL18 and BSK3/BSK6 were detected via bimolecular fluorescence complementation (BiFC) assays. YFP, YFP signal; Bright, bright field; Merge, merged image of YFP signal with bright field. Scale bars represent 50 μm. **c, d** Interactions between RTFL18 and BSK3/BSK6 were detected via semi-in vivo pull-down assay. **e** Interaction between RTFL18 and BSK3/BSK6 were detected via in vitro pull-down assay. **f** Interactions between Flag-RTFL18 and BSK3/6-GFP were detected via coimmunoprecipitation (CoIP) assay in *N. benthamiana* leaf cells. **g** Interactions between Flag-RTFL18 and BSK3/BSK6-GFP were detected via coimmunoprecipitation (CoIP) assay in Arabidopsis. **h** The BSK3/RTFL18 complex was predicted with AlphaFold-Multimer. The N-terminal domain (NTD) and C-terminal domain (CTD) of BSK3 are green yellow and cyan, respectively. RTFL18 is green. **i** The hydrophobic interactions between BSK3 and RTFL18. **j** Hydrogen bond interactions between BSK3 and RTFL18. **k** The hydrogen bond and electrostatic interactions between BSK3 and RTFL18. **l** Interactions between different forms of RTFL18 and BSK3 were detected via in vitro pull-down assays. 6His-RTFL18 WT (wild type), M1 (I39A, M43A, L47A), M2 (I40A, H50A, D51A), and GST-BSK3 were expressed in and purified from *E. coli*. Anti-GST antibodies were used to pull down GST-BSK3. Both the immunoprecipitated fractions and inputs were analyzed by immunoblotting with an anti-His and anti-GST antibodies, respectively. **m** Interactions between different forms of BSK3 and RTFL18 were detected via in vitro pull-down assays. GST-BSK3 WT (wild type), M1 (E368A, K385A, R393A), M2 (W374A, M378A, L382A), M3 (E368A, K385A, R393A, T374A, M378A, L382A) and His-RTFL18 were expressed in and purified from *E. coli*. Anti-His antibodies were used to pull down His-RTFL18. In **b**–**g** and **l, m**, each experiment was repeated three times with similar results. Source data are provided as a Source Data file.

RTFL18 and N367 and E368 of BSK3 (Fig. 4j). The C-terminal H50 and D51 residues of RTFL18 could form H-bonds and electrostatic interactions with E355 and R393 of BSK3, respectively (Fig. 4k). The complex formation can be further stabilized by the H-bond interactions between the side chain of BSK3 K385 and the main chains of RTFL18 L46 and L47. Similar to BSK3, BSK6 can also interact with RTFL18 (Supplementary Fig. 4d, e). Sequence alignment showed that the main amino acids of BSK3/6 involved in the interaction with RTFL18 are highly conserved (Supplementary Fig. 4f). RTFL21 might also interact with BSK3/6 based on structural superposition (Supplementary Fig. 4g, h). The main amino acids in RTFL18/21 in the interaction with BSKs are also highly conserved in RTFL family members (Supplementary Fig. 4i). To investigate the importance of these sites for their interactions, we expressed and purified recombinant RTFL18 M1 (I39A, M43A, L47A) and RTFL18 M2 (I40A, H50A, D51A), and mutated BSK3 M1 (E368A, K385A, R393A), BSK3 M2 (W374A, M378A, L382A), and BSK3 M3 (E368A, K385A, R393A, T374A, M378A, L382A) from *E. coli*. In pull-down assays, faint bands indicated reduced affinity of RTFL18 M1 and RTFL18 M2 with BSK3 (Fig. 4l). Mutated BSK3 also showed an impact on their binding (Fig. 4m). These results suggest that BSKs might share similar mechanisms in RTFL recognition and that these RTFL functions are probably redundant through BSKs.

## RTFL18 represses BSK3/6-mediated BR signaling

After confirming the expression of *BSK3/6* in the petiole (Supplementary Fig. 5a, b), we ordered T-DNA insertion mutants *bsk3* (*SALK_096500*) and *bsk6* (*SALK_104506*) (Supplementary Fig. 5c), and generated *BSK3* and *BSK6* overexpression lines (*35 S::BSK3-Flag* and *35 S::BSK6-Flag*) to investigate whether BSK3 and BSK6 participated in the shade avoidance response. The *bsk3* and *bsk6* mutants showed reduced cotyledon petiole and the first and second leaf petiole lengths, while the *BSK3* and *BSK6* overexpression lines displayed an opposite phenotype compared with the loss-of-function mutants under low R:FR conditions, suggesting that BSK3 and BSK6 positively regulate low R:FR-induced cotyledon petiole and the first and second leaf petiole elongation (Fig. 5a–c, Supplementary Fig. 5d). To further test whether RTFL18 genetically interacts with BSKs, we generated *dvl1-1D*BSK3ox* and *dvl1-1D*BSK6ox* lines, and these lines displayed longer cotyledon petiole and the first and second leaf petiole lengths than *dvl1-1D* (Fig. 5d–f and Supplementary Fig. 5e), indicating that RTFL18 negatively regulates the function of BSKs.

We found that overexpression of *RTFL18* did not affect the expression levels of *BSKs* (Supplementary Fig. 5f) and did not cause a negative effect on the protein level of BSK3/6 (Supplementary Fig. 5g, h). Given that intact BR signaling pathways are necessary for SAS, we examined the sensitivity of *dvl1-1D*, *35 S::RTFL18* and *rtfl18*rtfl21* to treatment with eBL and bikinin (a chemical inhibitor of GSK3-like kinases)[33]. Compared with the Col-0 line, lines overexpressing *RTFL18* were hyposensitive to eBL-induced cotyledon petiole elongation, similar to the responses of *bsk3* and *bri1-301* (a

mutant of BRI1), but *rtfl18*rtfl21* lines did not display a significantly enhanced response (Fig. 5g). Moreover, bikinin-induced cotyledon petiole elongation was enhanced in the *RTFL18ox*, *bsk3*, and *bri1-301* lines (Fig. 5h). These results indicate that RTFL18 negatively regulates BR signaling.

## RTFL18 antagonizes the stabilization of PIF4 under low R:FR conditions

PIF4 has been reported to be a target of BIN2, a GSK3-like kinase that functions as a core regulator in BR signaling, thereby facilitating the phosphorylation and degradation of PIF4[16]. Since RTFL18 negatively regulated BR signaling, we wondered whether the protein stability of PIF4 is also regulated by RTFL18. Using an endogenous antibody against PIF4, we found that the protein level of PIF4 was decreased significantly in *dvl1-1D* lines but increased in the *rtfl18*rtfl21* double-mutant line after low R:FR treatment (Fig. 6a and Supplementary Fig. 6a). In addition, the level of PIF4 protein was lower in the *bsk3* and *bsk6* mutants and higher in the *BSK3*- and *BSK6*-overexpressing lines (Fig. 6b, c and Supplementary Fig. 6b, c). Moreover, *BSK3/6* overexpression rescued the PIF4 protein level in the *dvl1-1D* background in response to low R:FR treatment (Fig. 6d, e and Supplementary Fig. 6d, e), suggesting that RTFL18 affected the protein level of PIF4 through BSK3/6 action. Continued overexpression of *PIF4* re-established the petiole and hypocotyl phenotype of the *dvl1-1D* line under white light and low R:FR light (Fig. 6f–h, Supplementary Fig. 6f). These results suggest that RTFL18 interacts with BSK3/6 to negatively regulate the protein level of PIF4 and mediates low R:FR-induced petiole elongation.

## RTFL18-BSK3/6-PIF4 regulates the expression of low R:FR responsive genes

To investigate the roles of RTFL18 and BSK3/6 in low R:FR-responsive transcription, we performed RNA-seq analysis in *dvl1-1D* and *bsk3bsk6* (*bsk36*) mutants and compared them with *pif4pif7* (*pif47*). After 6 h of low R:FR stimuli, 743 low R:FR responsive genes were identified in Col-0 seedlings (fold change >2 and *P* < 0.01, Supplementary Data 1). Through Gene Ontology (GO) analysis of these genes, "Response to auxin", "Regulation of growth", "Shade avoidance", and "Brassinosteroid homeostasis" terms were enriched (Supplementary Fig. 7a and Supplementary Data 2). The effects of low R:FR on these genes were compromised in *dvl1-1D*, *bsk36* and *pif47* (Fig. 7a, b). Specifically, 483, 455, and 601 genes showed changed low R:FR effects in *dvl1-1D*, *bsk36* and *pif47*, respectively (Fig. 7c). Hereinafter, these genes are called RTFL18-regulated, BSK3/6-regulated and PIF4/7-regulated low R:FR-responsive genes, respectively. There were 370 common genes among these gene clusters (Supplementary Data 3), in which "Signal transduction", "Regulation of growth", and "Brassinosteroid homeostasis" GO terms were enriched (Fig. 7d and Supplementary Data 4). We also compared the effects of low R:FR effects on BR-responsive genes and low R:FR-regulated genes in *dvl1-1D*, *bsk36* and *pif47* mutants (Supplementary Fig. 7b, c and Supplementary Data 5). As expected, the extent of expression

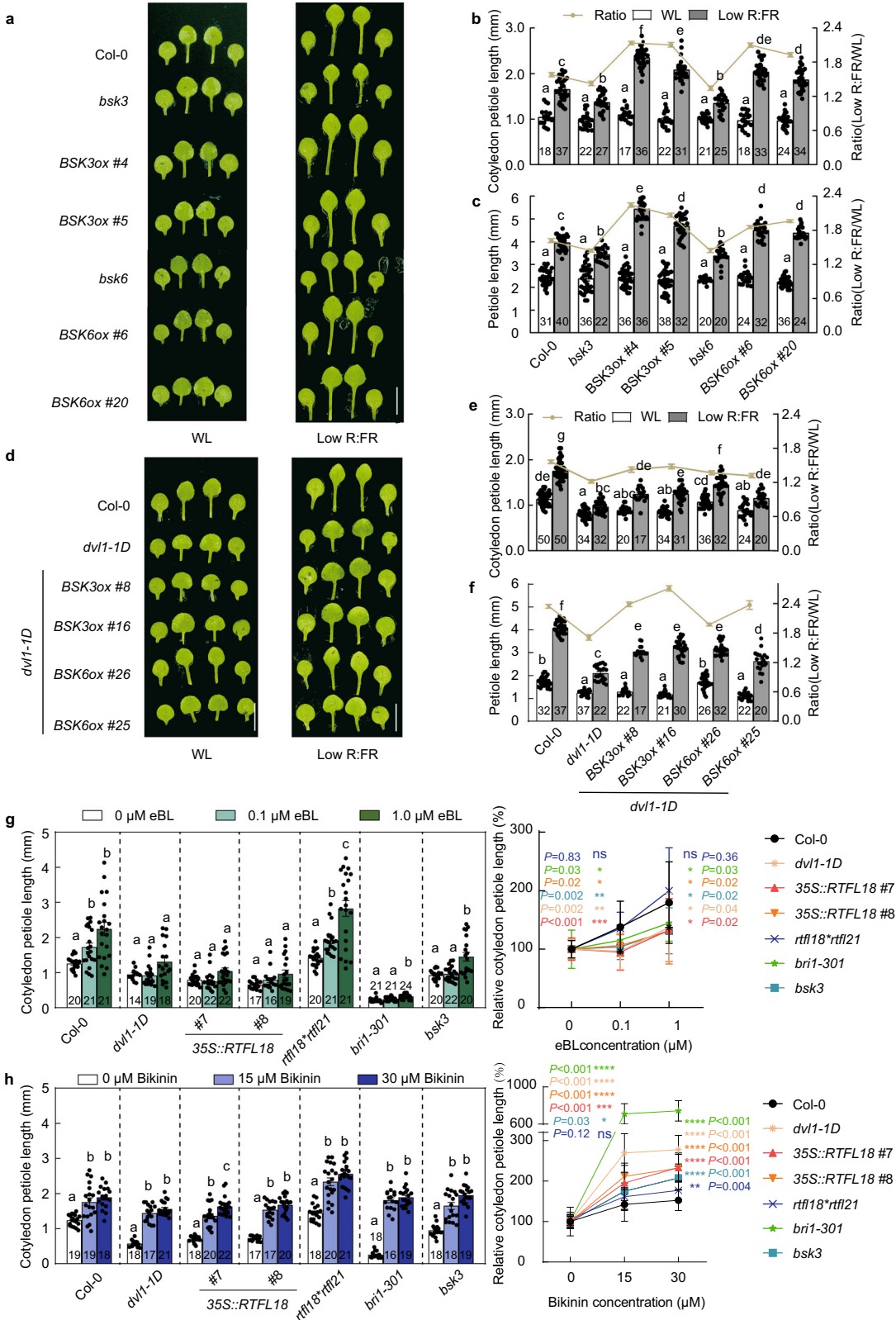

changes caused by low R:FR on these BR-responsive genes significantly decreased in the three mutants (Supplementary Fig. 7d), indicating that RTFL18-BSK3/6-PIF4 interfered with BR signaling under low R:FR. The expression levels of *HAT2*, *BAS1*, and *PAR1* were verified by RT–qPCR analysis (Fig. 7e). Based on the RNA-seq and RT–qPCR data, we conclude that RTFL18, BSK3/6 and PIF4/7 can regulate a substantial common subset of low R:FR responsive genes.

## Discussion

Plants have evolved complex signaling networks to sense environmental changes and adapt to environmental stresses. After exposure to vegetative shade, plants initiate the SAS to overcome shade conditions and prevent exaggerated growth responses. In this study, we found that the transcription of several members of the RTFL/DVL family is induced by low R:FR light and that these peptides were

**Fig. 5 | RTFL18 represses BSK3/6-mediated BR signaling. a** Pictures of Col-0, *bsk3*, *bsk6*, *35 S::BSK3ox-Flag* (*#4*, *#5*) (*BSK3ox #4*, *#5*), and *35 S::BSK6ox-Flag* (*#6*, *#20*) (*BSK6ox #6*, *#20*) seedlings grown under white light (WL) and low R:FR conditions. **b** The petiole lengths of the cotyledons in Col-0, *bsk3*, *bsk6*, *BSK3ox* (*#4*, *#5*), and *BSK6ox* (*#6*, *#20*) lines. **c** The petiole lengths of the first and second leaves in the Col-0, *bsk3*, *bsk6*, *BSK3ox* (*#4*, *#5*), and *BSK6ox* (*#6*, *#20*) lines. **d** Pictures of Col-0, *dvl1-1D*, *dvl1-1D 35 S::BSK3ox-Flag* (*#8*, *#16*) (*dvl1-1D*BSK3ox #8*, *#16*) and *dvl1-1D*35 S::BSK6ox-Flag* (*#26*, *#25*) (*dvl1-1D*BSK6ox #26*, *#25*) seedlings grown under WL and low R:FR conditions. **e** The petiole lengths of the cotyledons in the Col-0, *dvl1-1D*, *dvl1-1D*BSK3ox* (*#8*, *#16*), and *dvl1-1D*BSK6ox* (*#26*, *#25*) seedlings. **f** The petiole lengths of the first and second leaves in the Col-0, *dvl1-1D*, *dvl1-1D*BSK3ox* (*#8*, *#16*), and *dvl1-1D*BSK6ox* (*#26*, *#25*) seedlings. **g**, **h** The petiole lengths of the

cotyledons in the Col-0, *dvl1-1D*, *35 S::RTFL18* (*#7*, *#8*), *rtfl18*rtfl21*, *bri1-301*, and *bsk3* lines treated with 0, 0.1, or 1.0 μM eBL (**g**) or 0, 15, or 30 μM bikinin (**h**). The left panel represents the lengths and the right panel represents the response to eBL (**g**) or bikinin (**h**). In the right panel, the cotyledon petiole lengths in seedlings treated with 0 μM eBL (**g**) or 0 μM bikinin (**h**) were standardized to be "100%". The asterisks indicate significant differences to Col-0 (Student's two-sided *t* test, *\*P* < 0.05, *\*\*P* < 0.01, *\*\*\*P* < 0.001, ns indicates no significance). The numbers in the bar charts represent *n* of each sample. Data are presented as mean values +/− SEM. Letters in **b**, **c**, **e**–**h** indicate significant differences between mean values (two-way ANOVA: Tukey's multiple comparisons test, *P* < 0.01), and groups with the same letters are not significantly different. Scale bars represent 5 mm. Source data are provided as a Source Data file.

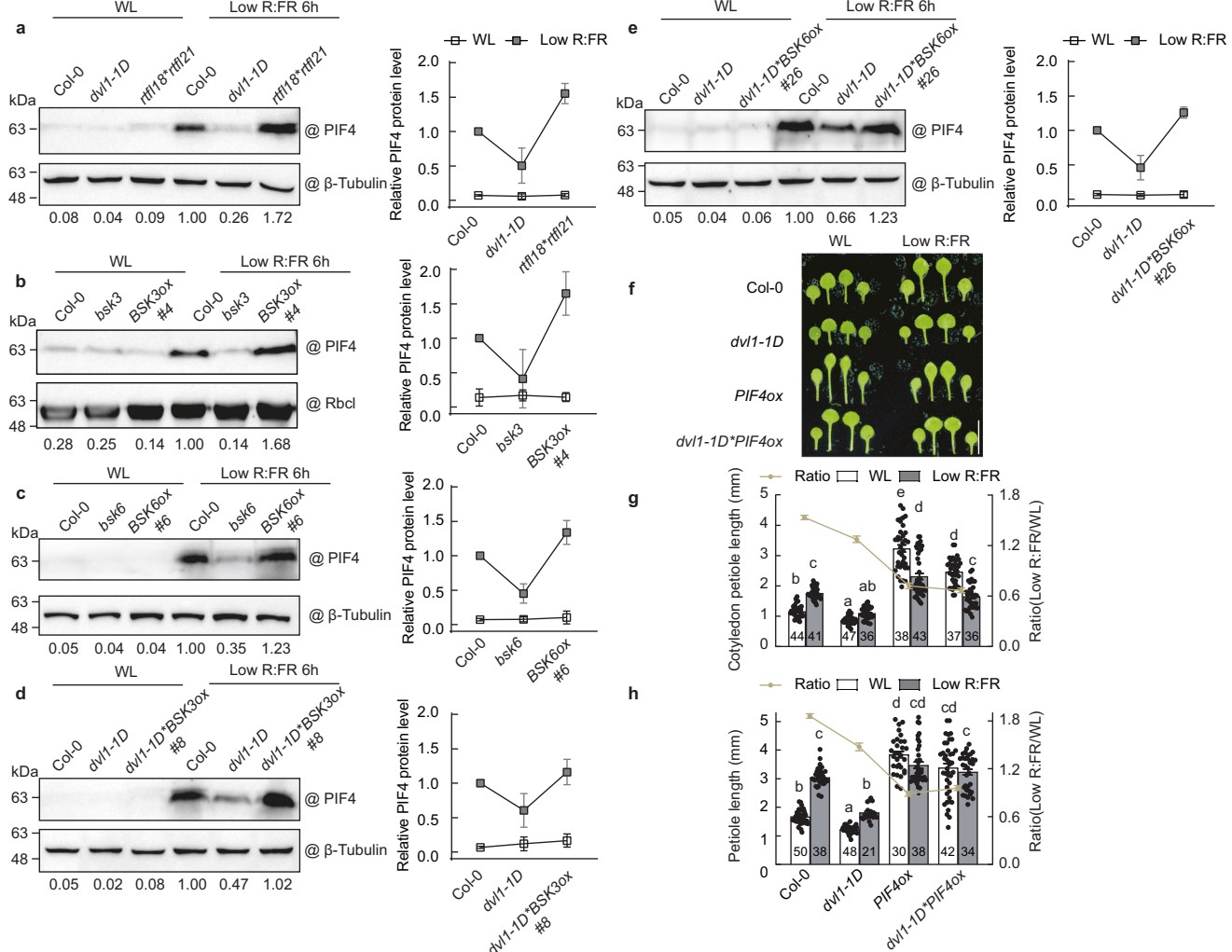

**Fig. 6 | RTFL18 antagonizes the stabilization of PIF4 under low R:FR conditions. a** The PIF4 protein levels in Col-0, *dvl1-1D*, and *rtfl18*rtfl21* lines grown under white light (WL) and low R:FR conditions. **b, c** The PIF4 protein levels in the Col-0, *bsk3*, *BSK3ox #4*, *bsk6*, and *BSK6ox #6* lines under WL and low R:FR conditions. **d, e** The PIF4 protein levels in the Col-0, *dvl1-1D*, *dvl1-1D*BSK3ox #8*, and *dvl1-1D*BSK6ox #26* lines under WL and low R:FR conditions. **f** Pictures of the Col-0, *dvl1-1D*, *PIF4ox*, and *dvl1-1D*PIF4ox* lines under WL and low R:FR conditions. **g** The petiole lengths of the cotyledons in the Col-0, *dvl1-1D*, *PIF4ox*, and *dvl1-1D*PIF4ox* lines grown under WL and low R:FR conditions. **h** The petiole lengths of the first and second leaves in the Col-0, *dvl1-1D*, *PIF4ox*, and *dvl1-1D*PIF4ox* lines grown under WL and low R:FR. In **a**–**e**, the right panels show the quantification analysis of three biological replicates

of PIF4 protein levels after normalization to β-tubulin/ Rbcl. Data are presented as mean values +/− SD (*n* = 3, *n* refers to biological replicates). The PIF4 protein levels of Col-0 under low R:FR were set as 1. Whole seedlings were used for experiments. Western blots were probed with an anti-PIF4 antibody. The levels of β-tubulin/ Rbcl indicate the loading concentration. In **g** and **h**, the numbers in the bar charts represent *n* of each sample. Data are presented as mean values +/− SEM. Letters indicate significant differences between mean values (two-way ANOVA: Tukey's multiple comparisons test, *P* < 0.01), and groups with the same letters are not significantly different. Scale bars represent 5 mm. Source data are provided as a Source Data file.

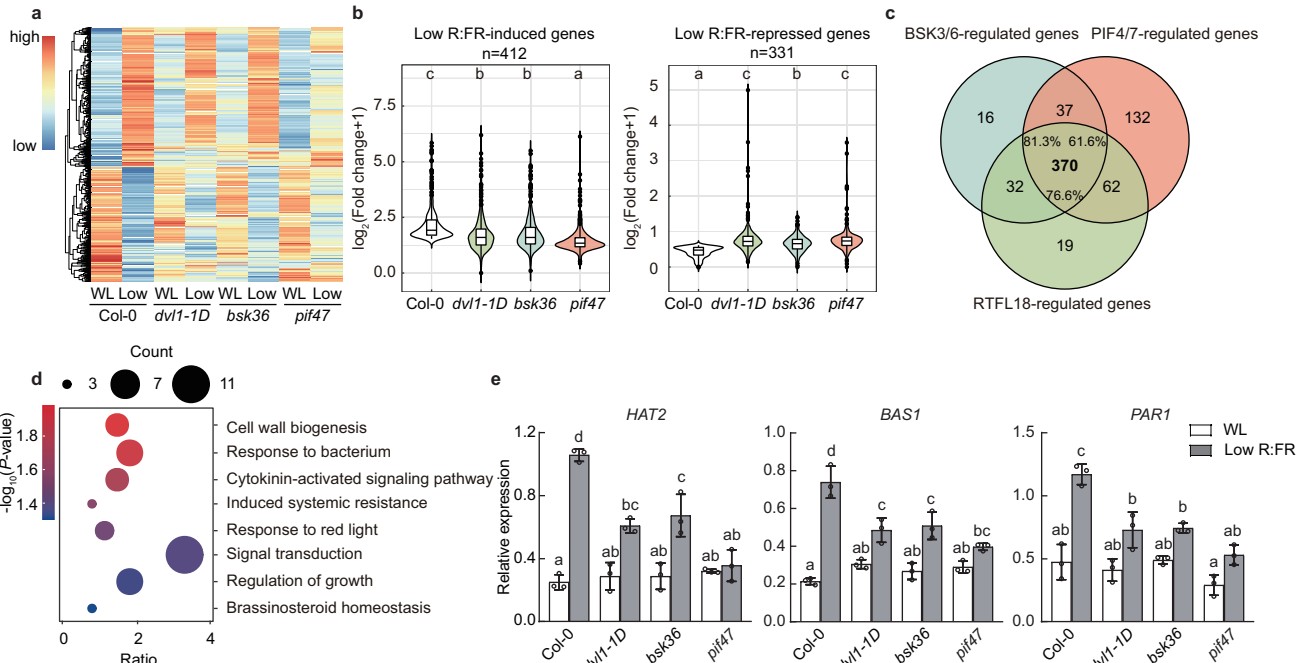

**Fig. 7 | RTFL18-BSK3/6-PIF4 regulates the expression of low R:FR-responsive genes. a** Heatmap showing the transcript levels of 743 low R:FR-regulated genes in Col-0, *dvl1-1D*, *bsk36*, and *pif47* under white light (WL) and low R:FR (Low) conditions. Red, yellow and blue rows indicate RNA expression at high, medium and low levels, respectively. **b** Boxplot representing the fold changes of low R:FR-regulated genes in Col-0, *dvl1-1D*, *bsk36*, and *pif47*, which was determined by comparing the transcript levels between WL and low R:FR conditions. Boxplots show the median (horizontal line), second to third quartiles (box), and whiskers extend to a maximum of 1.5 × interquartile range beyond the box. **c** Venn diagram representing 370 RTFL18-BSK3/6-PIF4/7 coregulated genes. **d** Gene Ontology (GO)

analysis of 370 RTFL18-BSK3/6-PIF4/7 coregulated genes. The size of each point represents the number of genes, and the color represents the *P* value, Fisher's one-tailed test (*P* < 0.05). **e** The relative expression levels of *HAT2*, *BAS1*, and *PAR1* under WL and low R:FR conditions were measured by RT–qPCR. The expression levels of genes were normalized against the expression of the reference gene *AT2G39960*. Data are presented as mean values +/− SD (*n* = 3, *n* refers to biological replicates). Letters (**b**, **e**) indicate significant differences between groups (**b** one-way ANOVA: Tukey's multiple comparisons test, *P* < 0.01; **e** two-way ANOVA: Tukey's multiple comparisons test, *P* < 0.01), and groups with the same letters are not significantly different. Source data are provided as a Source Data file.

involved in the inhibition of the SAS. RTFL18 interacts with BSK3/BSK6 and represses BR signaling, thereby reducing low R:FR-stabilized PIF4 levels and low R:FR-induced petiole elongation (Supplementary Fig. 7e).

The transcription of at least five members of the RTFL/DVL family can be induced by low R:FR conditions. Overexpression experiments suggested that some members of the RTFL/DVL family function identically[27]. We observed a strong low R:FR-defective phenotype acquired in *dvl1-1D* and *RTFLs*-overexpressing lines, which manifested as reduced cotyledon petiole, first and second leaf petiole and hypocotyl elongation (Fig. 3a–c, Supplemental Fig. 3b–e). However, we are confident only in the results showing longer petioles in the *rtfl18*rtfl21* line (Fig. 3e–g). Functional redundancy among family members and tissue-specific expression patterns may explain these findings. *RTFL18* was expressed mainly in leaves and petioles (Fig. 2a, b). *RTFL21* was expressed mainly in leaves, stems, and flowers; *RTFL17* was expressed mainly in leaves; *RTFL19* was expressed mainly in stems; and *RTFL15* was expressed mainly in roots and flowers[27]. These expression patterns suggest that different RTFL/DVL peptides may function in different tissues. Plants carrying high-order mutants with genes expressed in similar patterns might exhibit a more extensively defective SAS. Also, functional interactions of RTFLs with putative interactors, such as BSKs, may not be similar between all tissues. The tissue expression patterns of BSK3 and BSK6 were shown in Supplementary Fig. 5a, b. The tissue specific expression levels of these putative interactors will also inference in vivo interactions. More tissue specific studies are required in future.

Brassinosteroid (BR) biosynthesis and signaling are required for normal SAS function. However, shade alters BR levels in a dynamic

fashion[34]. Although an increase in castasterone (CS, precursor of BR) content was detected after exposure to 45 min of shade, changes in the phosphorylation rate of BES1 seemed inconsistent with the CS content[35]. Organ-specific COP1 control of BES1 stability adjusts plant growth patterns under shade conditions[36]. Tissue-specific shade-induced negative regulators of BR signal transduction interfered with downstream signaling and negatively regulated BR levels under shade conditions. In the current study, RTFL18 negatively regulated the BR signaling pathway, as evidenced by its responses to eBL and bikinin treatment (Fig. 5g, h). Low R:FR-induced *RTFL18* expression might contribute to the dynamic changes in BR signaling in specific tissues under low R:FR conditions. It may take a time lag from the induction of RTFL18 to the effect on BR signaling, the changes of PIF4 protein level and then the final phenotypes. The working time of RTFLs-BSKs module could be at the same time with or be later than the activation time of SAS. The existence of time-lag enables plants to have enough time to accurately perceive changes in the environment and then conduct the precise regulations. Plants rely on a variety of long-term (plant hormones) and short-term (peptides and membrane proteins) communication mechanisms to coordinate their growth under low R:FR conditions.

In Arabidopsis, there are 12 BSK proteins that carry putative kinase catalytic domains in the N-terminus and harbor tetra-tricopeptide repeats (TPRs) in the C-terminus. Several BSKs have been reported to play a partially overlapping role in BR signaling[32]. We analyzed the interactions of RTFL18 with BSK3 and BSK6 and identified *bsk3*- and *bsk6*-induced phenotype changes under low R:FR conditions. BSK3 and BSK6 are positive regulators of low R:FR responses. Previous studies have shown that PIF4, PIF5, and PIF7

redundantly promote the transcription of *BSK5* by binding directly to the G-box of the *BSK5* promoter under shade light, which leads to the activation of BES1[17]. Among these BSKs, only the expression of *BSK5* was induced by shade and by PIFs[17]. RTFL18 did not affect the expression levels of *BSK3/6* (Supplementary Fig. 5f). Low R:FR-induced *RTFL18* expression changed the activity levels of BSKs through direct binding, which might have influenced the interactions between BSKs and their downstream factors in BR signaling to ultimately suppress BR signaling.

The protein stability of PIF4 was influenced by RTFL18 and BSK3/6 (Fig. 6a–c). Among RTFL18 interacting factors identified through an IP-MS analysis, we found MAPKK5 and MAPK3/6 (Supplementary Fig. 4b). Arabidopsis BSK1 directly interacts with the immune receptor FLAGELLIN SENSING2 (FLS2) and further phosphorylates MAPKKK5, thereby activating pattern-triggered immunity (PTI)[37,38]. In addition, the MAPKKK10-MAPK6 pathway has been shown to mediate red light-regulated cotyledon opening, which is mediated through phosphorylation of PIF3 in Arabidopsis[39]. Therefore, we speculated that the MAPKKK-MAPKK-MAPK cascade might also act downstream of RTFL18-BSK3/6 and affect the stabilization of PIF4 under shade conditions. The MAPKK5 and MAPK3/6 cascades have been reported to be involved in responses to multiple environmental stimuli. For instance, MPK3/6 are activated by cold[40], heat[41], and salt[42]. However, the effects of shade conditions remain to be determined with evidence, including data on the effects of shade light on the activity of MAPKKs and MAPKs and the protein levels of PIF4 in *MAPKK*- and *MAPK*-mutants.

Taken together, our work characterizes the small peptide RTFL18 and reveals BSK3/6 to be RTFL18-interacting factors. Low R:FR-induced RTFL expression represses BR signaling and reduces low R:FR-stabilized PIF4 levels, which results in defective low R:FR-induced petiole elongation. The findings reveal that a negative mechanism mediated by RTFLs occurs to prevent petioles from excessively elongating under low R:FR conditions. Our results highlight a discovered and important function of non-secreted peptides that engage in the crosstalk between BR signaling and SAS components.

## Methods

### Plant materials

The wild-type and all mutants used in this study had an Arabidopsis Columbia-0 (Col-0) ecotype background. The *phyB-9^BC^* (with mutation in *PHYB*, but without the mutation in *VEN4*)[43], *pif7-1*[7], *pif7-2*[7], and *pifq*[7] mutants were described previously. The *rtfl18, rtfl21*, and *rtfl18*rtfl21* mutants were generated using CRISPR/Cas9 technology by the Biorun Biological Company. Single guide RNAs of *RTFL18* or *RTFL21* were cloned into pCAMBIA1300-pYAO:Cas9 vectors. These plasmid constructs were introduced into *Rhizobium radiobacter GV3101* and transformed by floral infiltration into Arabidopsis wild-type (Col-0) plants. T1 transgenic plants were selected via hygromycin resistance, and their identity was confirmed by DNA sequencing. Homozygous T3 generation seeds were selected for further analysis. The lines with T-DNA insertion of the *bsk3* mutant and *bsk6* mutant were obtained from AraShare (http://www.arashare.cn), and their identities were verified by PCR genotyping.

To obtain transgenic lines overexpressing the *RTFL18* gene, the coding region of the *RTFL18* gene was amplified by PCR and cloned into pCAMBIA1306 harboring a 3 × Flag tag under the control of the Cauliflower mosaic virus (CaMV) 35 S promoter to generate *35 S::RTFL18-Flag, 35 S::Flag-RTFL18* and *35 S::RTFL18*. To obtain transgenic lines overexpressing the *BSK3* and *BSK6* genes, the coding regions of the *BSK3* and *BSK6* genes were amplified by PCR and cloned into pCAMBIA1306 harboring a 3 × Flag tag under the control of the CaMV 35 S promoter to generate *35 S::BSK3-Flag and 35 S::BSK6-Flag*, respectively. To obtain *Flag-RTFL18*BSK3-GFP* and *Flag-RTFL18*BSK6-GFP* lines, the coding regions of the *BSK3* and *BSK6* genes were amplified by PCR and cloned into pCAMBIA2302 to generate *35 S::BSK3-GFP* and *35 S::BSK6-*

*GFP*, respectively. The resulting plasmids were introduced into *Rhizobium radiobacter GV3101* and transformed by floral infiltration into wild-type (Col-0) or *35 S:: Flag-RTFL18* plants. Transgenic plants were screened on half-strength Murashige and Skoog (1/2 MS) nutrient medium containing the appropriate antibiotics, and their identities were confirmed by immunoblot analysis. *35 S::PIF7-Flash* and *35 S::PIF4-Flash* were described previously[7,44]. The lines with dual overexpression to generate *dvl1-1D*35 S::BSK3-Flag, dvl1-1D*35 S::BSK6-Flag* and *dvl1-1D*35 S::PIF4-Flash* were prepared by genetic crossing and identified by PCR genotyping, RT–qPCR and western blotting.

### Growth conditions and phenotype analysis

For a phenotype analysis of petiole lengths of the cotyledons and the first and second leaves, seeds were sown on 1/2 Murashige and Skoog (MS) medium (2.3 g/L MS, 1.2% Agar). After stratification for 3 days, the seedlings were transferred into a growth chamber (JIUPO, BPC500-2H) with white light (LED light, JIUPO, JIUPO-5050TLED-300-S, R: ~20 μmol m$^{-2}$ s$^{-1}$, B: ~20 μmol m$^{-2}$ s$^{-1}$, FR: ~5 μmol m$^{-2}$ s$^{-1}$), 22 °C. For low R:FR treatment, seedlings grown for three days in white light were placed either under white light (R: ~20 μmol m$^{-2}$ s$^{-1}$, B: ~20 μmol m$^{-2}$ s$^{-1}$, FR: ~5 μmol m$^{-2}$ s$^{-1}$) or transferred to low R:FR (LED light, JIUPO, JIUPO-5050TLED-300-S, R: ~20 μmol m$^{-2}$ s$^{-1}$, B: ~20 μmol m$^{-2}$ s$^{-1}$, FR: ~60 μmol m$^{-2}$ s$^{-1}$) for 8 days. The cotyledon and the first and second leaf petiole lengths were measured and analyzed.

For a phenotype analysis of hypocotyl length, seeds were sown on 1/2 Murashige and Skoog (MS) medium (2.3 g/L MS, 1% Agar). After stratification for 3 days, the seedlings were transferred into a growth chamber (JIUPO, BPC500-2H) with white light (LED light, JIUPO, JIUPO-5050TLED-300-S, R: ~20 μmol m$^{-2}$ s$^{-1}$, B: ~20 μmol m$^{-2}$ s$^{-1}$, FR: ~5 μmol m$^{-2}$ s$^{-1}$), 22 °C. For low R:FR treatment, seedlings grown for three days in white light were placed either under white light (R: ~20 μmol m$^{-2}$ s$^{-1}$, B: ~20 μmol m$^{-2}$ s$^{-1}$, FR: ~5 μmol m$^{-2}$ s$^{-1}$) or transferred to low R:FR (R: ~20 μmol m$^{-2}$ s$^{-1}$, B: ~20 μmol m$^{-2}$ s$^{-1}$, FR: ~60 μmol m$^{-2}$ s$^{-1}$) for 4 days. The hypocotyl lengths were measured and analyzed.

### Subcellular localization analysis

For subcellular localization analysis with *Nicotiana benthamiana*, *35 S::GFP-RTFL18 and 35 S::GFP* plasmids were transferred into *Rhizobium radiobacter GV3101* and infiltrated into tobacco leaves using a needleless syringe. After infiltration, plants were grown in the dark for 24 h and then with a 16 h light/8 h dark photoperiod for 48–60 h. YFP fluorescence was imaged under a confocal laser scanning microscope (OLYMPUS FV3000).

### Chromatin immunoprecipitation (ChIP) assay

In Fig. 1h, seedlings of *PIF7ox* (*35 S::PIF7-Flash(9 × Myc-6 × His-3 × Flag)*) and *PIF4ox* (*35 S::PIF4-Flash(9 × Myc-6 × His-3 × Flag)*) were grown under continuous white light (R ~ 20 μmol·m$^{-2}$·s$^{-1}$, B ~ 20 μmol·m$^{-2}$·s$^{-1}$, FR ~ 5 μmol·m$^{-2}$·s$^{-1}$) for 10 days and treated with low R:FR or continued to be grown in white light for 1 h. Whole seedlings were harvested and crosslinked in a prechilled crosslinking buffer (10 mM Tris-HCl, 0.4 M sucrose, 1 mM EDTA, 0.2 × protease inhibitor cocktail, 1 mM PMSF, and 1% formaldehyde pH 8.0) for 15 min in a vacuum. Glycine (125 mM) was used to quench the crosslinking performed in a vacuum after 7 min of reaction and then seedlings were washed four times in double-distilled water. The seedlings were frozen and ground into powder with liquid nitrogen for chromatin extraction. More than 1.0 g seedlings were ground and nucleus were extracted using lysis buffer [50 mM HEPES pH 7.5, 150 mM NaCl, 1 mM EDTA, 1% Triton X-100, 5 mM β-mercaptoethano, 10% glycerol, and protease inhibitor cocktail (Roche, 4693132001)]. A Bioruptor was used at high power in 30-s on/30-s off cycles to achieve an average chromatin size of approximately 300 bp using lysis buffer with 0.5% SDS. The fragmented chromatin was incubated with an anti-Myc antibody (GNI, GNI4410-MC) or anti-Flag antibody (GNI, GNI4410-FG) overnight at 4 °C. rProtein A beads (Smart-

lifesciences, SA012025) were then incubated with the mixture for approximately 1 h at 4 °C. The complex was eluted with elution buffer (1% SDS, 0.1 M NaHCO3) after gradual washes successively by low salt wash buffer [20 mM Tris-HCl (pH 8.0), 2 mM EDTA, 150 mM NaCl, 0.1% SDS, 1% Triton X-100], high salt wash buffer [20 mM Tris-HCl (pH 8.0), 2 mM EDTA, 500 mM NaCl, 0.1%SDS, 1% Triton X-100], LiCl wash buffer [10 mM Tris-HCl (pH 8.0), 1 mM EDTA, 1% sodium deoxycholate, 250 mM LiCl, 1% NP-40] and TE buffer [10 mM Tris-HCl (pH 8.0), 1 mM EDTA]. The sample with 5 M NaCl treated overnight at 65 °C. Subsequently, DNA purification is performed. The IP efficiency was determined as $[2\hat{\ }(Ct_{input} \text{ average} - Ct_{sample})]/(5 \times 100)$.

## Immunofluorescence

In Fig. 2e and Supplementary Fig. 2c, six-day-old *35 S::Flag-RTFL18* seedlings grown in WL were fixed in FAA solution (Gefanbio, M072). Seedlings were washed by 1 × PBS. Add enzymatic solution (10 mg/ml cellulase R-10, 2.5 mg/ml Macerozyme R-10, 0.4 M Mannitol, 20 mM MES pH = 5.7, 20 mM KCl, 10 mM CaCl₂) and shake for 2 h. Seedlings were washed by 1 × PBS. Seedlings were treated for 1 h in a 5% BAS solution containing 50 mM glycine. Flag-RTFL18 was detected via anti-Flag (GNI, GNI4110-FG) antibody as a primary antibody and Alexa Fluor 488-labeled Goat Anti-Mouse IgG antibodies as a secondary antibody (Beyotime, A0428). Fluorescence was imaged under a confocal laser scanning microscope (OLYMPUS FV3000).

## Luciferase complementation imaging (LCI) assay

The coding sequences of the *RTFL18*, *RTFL21*, *BSK3*, and *BSK6* genes were cloned into the N-terminal half (NLUC) or C-terminal half (CLUC) of the LCI vector to obtain cLuc-RTFL18, cLuc-RTFL21, BSK3-nLuc and BSK6-nLuc. The resulting plasmids were transferred into *Rhizobium radiobacter GV3101* and infiltrated into young but fully expanded leaves of 7-week-old tobacco (*Nicotiana benthamiana*) leaves using a needleless syringe. After infiltration, the plants were grown in the dark for 24 h and with a 16 h light/8 h dark photoperiod for 48–60 h. Luc activity was observed with a CCD imaging apparatus.

## Bimolecular fluorescent complementation (BiFC) assay

Yellow fluorescent protein (YFP) was used in the BiFC assays. The coding regions in the *RTFL18*, *BSK3*, and *BSK6* genes were cloned into the pXY106, pXY104, and pXY104 binary vectors to generate nYFP-RTFL18, BSK3-cYFP and BSK6-cYFP, respectively. The resulting plasmids were transferred into *Rhizobium radiobacter GV3101* and infiltrated into tobacco leaves with buffer (10 mM MES, 1 mM AS, 10 mM MgCl₂) using a needleless syringe. After infiltration, plants were grown in the dark for 24 h and then with a 16 h light/8 h dark photoperiod for 48–60 h. YFP fluorescence was imaged under a confocal laser scanning microscope (OLYMPUS FV3000).

## Semi-in vivo pull-down assay

The full-length CDSs of RTFL18 were cloned into pGEX4T-1. The recombinant plasmids were transformed into *E. coli* strain Rosetta (DE3), and the proteins were induced by incubation in 200 ml of LB medium containing 0.5 mM isopropyl-β-D-thiogalactopyranoside (IPTG) at 16 °C for 16 h. The cells were harvested by centrifugation at 4000 × g and resuspended in 1X PBS. Then the GST and GST-RTFL18 were purified. *35 S::BSK3-Flag* and *35 S::BSK6-Flag* were transiently expressed in tobacco (*N. benthamiana*) leaves to obtain BSK3/BSK6-Flag. Tobacco leaves were ground into a powder with liquid nitrogen, and proteins were extracted with extraction buffer [100 mM Tris-HCl (pH 7.5), 300 mM NaCl, 2 mM EDTA, 0.1% Triton X-100, 10% glycerol, and 1 × protease inhibitor cocktail (Roche, 4693132001)]. The supernatant obtained after centrifugation at 15,700 g (4 °C) mixed with purified GST or GST-RTFL18 and incubated with prewashed glutathione-agarose beads (Smart-lifesciences, M00201) while rotating at 4 °C for 2 h. The beads were collected and washed six times with

buffer (1 × PBS containing 1 × protease inhibitor cocktail and 1 mM DTT) and resuspended in protein-loading buffer. The immunoprecipitates and inputs were separated on SDS-PAGE gels and immunoblotted with anti-GST antibody (Abmart, M20007) or anti-Flag antibody (GNI, GNI4110-FG).

## In vitro pull-down assay

The full-length coding regions of *RTFL18*, *BSK3*, and *BSK6* were cloned into a pET-SUMO vector and pGEX4-T-1 vector. The mutant forms of *RTFL18* and *BSK3* were cloned into pET-SUMO vector and pGEX4-T-1 vector. The constructs were transformed into *E. coli* BL21 (DB3) to produce Sumo-His-RTFL18, His-RTFL18 (WT, M1, M2), GST-BSK3 (WT, M1, M2, M3), GST-BSK6, and GST protein.

In Fig. 4e, Sumo-His-RTFL18 was incubated with a Ni-NTA 6FF (Smart-lifesciences, SA005025) beads at 4 °C for 2 h in 1 × PBS containing 1 × protease inhibitor cocktail and 1 mM DTT. The beads were collected and washed three times with incubation buffer on ice. Then, GST and GST-BSK3/BSK6 were incubated with the Ni beads at 4 °C for 2 h in 1 × PBS containing 1 × protease inhibitor cocktail and 1 mM DTT. The beads were collected and washed six times with incubation buffer and resuspended in 2 vols of the protein-loading buffer. The immunoprecipitates and inputs were separated on SDS-PAGE gels and immunoblotted with anti-GST (Abmart, M20007) or anti-His antibody (GNI, GNI4410-HS).

In Fig. 4l, purified GST-BSK3 WT were incubated with 4FF glutathione beads (Smart-lifesciences, SA010010) at 4 °C for 2 h in 1 × PBS containing 1 × protease inhibitor cocktail and 1 mM DTT. The beads were collected and washed three times with incubation buffer on ice. Then, the His-RTFL18 (WT, M1, M2) proteins were incubated with the Ni beads at 4 °C for 2 h in 1 × PBS containing 1 × protease inhibitor cocktail and 1 mM DTT, respectively. The beads were collected and washed 6 times with incubation buffer and resuspended in 2 vols of the protein-loading buffer. The immunoprecipitates and inputs were separated on SDS-PAGE gels and immunoblotted with anti-His antibody (GNI, GNI4410-HS) and anti-GST antibody (Abmart, M20007).

In Fig. 4m, purified His-RTFL18 WT were incubated with Ni-NTA 6FF (Smart-lifesciences, SA005025) beads at 4 °C for 2 h in 1 × PBS containing 1 × protease inhibitor cocktail and 1 mM DTT. The beads were collected and washed three times with incubation buffer on ice. Then, the GST-BSK3 (WT, M1, M2, M3) proteins were incubated with the Ni beads at 4 °C for 2 h in 1 × PBS containing 1 × protease inhibitor cocktail and 1 mM DTT, respectively. The beads were collected and washed 6 times with incubation buffer and resuspended in 2 vols of the protein-loading buffer. The immunoprecipitates and inputs were separated on SDS-PAGE gels and immunoblotted with anti-GST antibody (Abmart, M20007) or anti-His antibody (GNI, GNI4410-HS).

## Coimmunoprecipitation (CoIP) assay

In *N. benthamiana*, *35S::Flag-RTFL18* and *35 S:: GFP*, *35S::BSK3-GFP* or *35S ::BSK6-GFP* were transiently expressed in *N. benthamiana* leaves. Plants were grown in the dark for 24 h and with a 16 h light/8 h dark photoperiod for 48–60 h, then protein were extracted in lysis buffer [50 mM Tris-HCl pH 7, 150 mM NaCl, 5 mM DTT, 2% NP-40, 1% Triton X-100, 1 mM PMSF and 1× protease inhibitor cocktail (Roche, 4693132001)]. The crude protein extract was filtered through Miracloth, cleared by centrifugation (20 min, 10000 g). A fraction was conserved as input control. IP was carried out for 2 h at 4 °C using anti-GFP Affinity beads (Smart-Lifesciences, SA070001). Beads were washed 5 times for 5 min in lysis buffer. The beads were collected and resuspended in 2 vols of the protein-loading buffer. The immunoprecipitates and inputs were separated on SDS-PAGE gels and immunoblotted with anti-Flag antibody (GNI, GNI4110-FG) and anti-GFP antibody (Abmart, M20004).

In Arabidopsis, total protein extracts were prepared from *35 S::Flag-RTFL18*35 S::BSK3-GFP* and *35 S::Flag-RTFL18*35 S::BSK6-GFP*

seedlings plants grown in duplicate under white light for 3 weeks and then one of the duplicates was treated with low R:FR for 1 h and the other of duplicate was maintained under white light for an additional hour. *35 S::Flag-RTFL18*35 S::BSK3-GFP*, *35 S::Flag-RTFL18*35 S::BSK6-GFP*, and *35 S::Flag-RTFL18* seedlings were ground in liquid nitrogen and proteins were extracted in lysis buffer [50 mM Tris-HCl pH 7, 150 mM NaCl, 5 mM DTT, 2% NP-40, 1% Triton X-100, 1 mM PMSF and 1 × protease inhibitor cocktail (Roche, 4693132001)]. The crude protein extract was filtered through Miracloth, cleared by centrifugation (20 min, 10000 g). A fraction was conserved as input control. IP was carried out for 2 h at 4 °C using anti-GFP Affinity beads (Smart-Lifesciences, SA070001). Beads were washed 3 times for 5 min in lysis buffer. Immunoprecipitated proteins were eluted by boiling, separated by SDS-PAGE and detected by Western blotting using anti-Flag antibody (GNI, GNI4410-FG) and anti-GFP antibody (Abmart, M20004).

## Modeling of the RTFL/BSK complex
To clarify how RTFLs interact with BSK family proteins, we built one BSK/RTFL model using AlphaFold-Multimer. One program was developed to precisely predict protein complexes. The AlphaFold-Multimer open-source code (https://gitcode.net/mirrors/deepmind/alphafold?utm_source=csdn_github_accelerator) was downloaded and installed on a local server. The complex model was generated by the program with all default settings. The model with the highest confidence score was utilized in the RTFL and BSK interaction analysis. All structures were displayed by the ChimeraX (https://www.rbvi.ucsf.edu/chimerax) program[45].

## RNA-seq library preparation, construction, and analysis
We referred to methods in the literature for RNA-seq[46] and the details are as follows. Six-day-old white light-grown seedlings were kept in white light or were transferred to low R:FR conditions for 6 h, and whole seedlings were collected. We sent 0.1–0.2 g fresh plant material to Majorbio company (https://www.majorbio.com/) to perform standard RNA-seq library preparation and construction. Three replicates were prepared for each genotype. Total RNA was extracted from the tissue using RNA Purification Reagent according the manufacturer's instructions and genomic DNA was removed using DNase I (TaKaRa). RNA degradation and contamination was monitored on 1% agarose gels. Then RNA quality was determined by 2100 Bioanalyser (Agilent Technologies) and quantified using the ND-2000 (NanoDrop Technologies). RNA purification, reverse transcription, library construction and sequencing were performed at Shanghai Majorbio Bio-pharm Biotechnology Co., Ltd. (Shanghai, China) according to the manufacturer's instructions (Illumina, San Diego, CA). To identify DEGs (differential expression genes) between two different samples, the expression level of each gene was calculated according to the Fragments Per Kilobases per Millionreads (FPKM) method. Differential expression analysis was performed using DESeq2[47], and DEGs with |log2 (fold change)| ≥1 and P ≤ 0.01 were considered to be significantly differentially expressed genes.

Gene Ontology enrichment analysis was performed by the website Plant Regulomics (http://bioinfo.sibs.ac.cn/plant-regulomics/). The intersection between two or three sets of genes was determined using Venny2.1 (https://bioinfogp.cnb.csic.es/tools/venny/index.html).

## Quantitative RT−PCR analysis
In Fig. 1b, f, and Supplementary Fig. 1c, seedlings grown for 5 days with white light were transferred to low R:FR or remained in white light for 1 h. In Fig. 1d, seedlings grown for 5 days in the dark were transferred to white light or remained in the dark and grown for 1 h. In Fig. 1e, seedlings were grown for 5 days under white light. In Fig. 2b and Supplementary Fig. 2b, seedlings grown for 10 days with white light were transferred to low R:FR conditions or continually exposed to white light for 1 h, and the different tissues were separated. In Fig. 7e,

seedlings grown for 6 days with white light were transferred to low R:FR conditions or continual white light exposure for 6 h.

Whole seedlings were used for experiments. Total RNA was extracted from seedlings via the TRIzol (Ambion, 15596018) RNA extraction method. A 2-μg aliquot of RNA was used for first-strand cDNA synthesis with a FastQuant RT kit (Tiangen, KR118-02). Analysis was performed with a Real-Time System CFX96™ C1000 Thermal Cycler (Bio-Rad). All experiments were repeated at least three times and independent biological experiments were repeated three times. The *AT2G39960* gene was used as the internal control. The expression levels of target genes were normalized against the expression of the reference gene *AT2G39960*. Data are presented as mean values +/− SD ($n = 3$, $n$ refers to biological replicates). The significant differences were calculated by Multiple $t$ tests (*$P < 0.05$, **$P < 0.01$, ***$P < 0.001$, ns indicates no significance), one-way ANOVA: Tukey's multiple comparisons test and two-way ANOVA: Tukey's multiple comparisons test ($P < 0.01$). Multiple $t$ tests were used 1% False Discovery Rate (FDR) approach with two-stage step-up method of Benjamini, Krieger, and Yekutieli. The primers used for RT-qPCR are listed in Supplementary Table 1.

## Western blotting and analysis
In Fig. 2c, seedlings of *35 S::Flag-RTFL18* grown for 6 days under white light were transferred to low R:FR conditions for 0, 30, 90, and 180 mins. In Fig. 6a–e and Supplementary Fig. 6a–e, seedlings grown for 6 days in white light were transferred to low R:FR conditions and grown for 6 h.

In Figs. 2, 3, Supplementary Fig. 5, Fig. 6, and Supplementary Fig. 6, whole seedlings were ground to a fine powder in liquid nitrogen. Total proteins were extracted with extraction buffer [125 mM Tris-HCl (pH 8.0), 375 mM NaCl, 2.5 mM EDTA, 1% SDS, and 1% b-mercaptoethanol]. The proteins were separated on SDS-PAGE gels followed by wet transfer to nitrocellulose membranes. The antibodies used in this study are described in the Antibody information. Signals were detected using a ShareBio ECL kit (ShareBio, sb-wb011).

We referred to methods in the literature to quantification analysis of western blot[48] and the details are as follows. Only use non-overexposed images for the quantitative analysis. ImageJ software was used to calculate the mean gray values of the bands. Convert the image to grayscale and rectangular selections tool from the ImageJ toolbar were used to select the target band. The software converted the band grayscale into a peak plot, and the area of the peak was the quantified value of the grayscale. The mean gray values of the target proteins were divided by the mean gray value of the internal reference to obtain the relative protein level of the target protein. The PIF4 protein levels of Col-0 under low R:FR were standardized as "1" after normalization.

In Fig. 2f, proteins in different cellular fractions were extracted from 7-day-old *35 S::Flag-RTFL18* transgenic seedlings using a Minute™ Plasma Membrane Protein isolation kit (MobiTec, SM-005-p-INV) and subjected to SDS-PAGE electrophoresis and immunoblot analysis; PM H⁺-ATPase was detected using Anti-H⁺-ATPase antibody (PHYTOAB, PHY2285A) nuclear H3 was detected using anti-H3 antibody (Abmart, P30266M), and cytoplasm Rbcl was detected using anti-Rbcl antibody (Agrisera, AS03037).

## Sensitivity to eBL and bikinin
We referred to methods in the literature[49,50] and made improvements. eBL (Aladdin, E128317-10mg) and bikinin (Sigma, SML0094-5 MG) were each dissolved to 100 mM in dimethyl sulfoxide as stock solutions and stored at −20 °C. These chemical compounds were added to 1/2 MS below 65 °C.

Surface-sterilized seeds were plated on 1/2 MS with 1.2% (w/v) agar and stratified at 4 °C for 3 d. Seedlings were grown for 5 d on 1/2 MS medium under white light. To examine the effects of eBL and bikinin, whole seedlings were transferred onto growth medium containing eBL

or bikinin at the indicated concentration (eBL: 0, 0.1, or 10 µM; bikinin, 0, 15, or 30 µM) for 3 d. Data measurement and analysis can be found in the following section.

## Quantitative measurement and analysis

We used a scanner (HP Scanjet 8270) to obtain images of seedlings. Cotyledon petiole lengths, first and second leaf petiole lengths and hypocotyl lengths of Fig. 3, Supplementary Fig. 3, Fig. 5, Supplementary Fig. 5, Fig. 6 and Supplementary Fig. 6 on scanned images are measured with ImageJ software (http://www.scioncorp.com).

Statistical parameters including the exact value of $n$, the definition of center, dispersion, and precision measures (mean ± SEM) and statistical significance are reported in the Figures and Figure Legends.

## Antibody information

The following antibodies were purchased: anti-Myc antibody (GNI, GNI4410-MC, 1:4000); anti-Flag antibody (GNI, GNI4410-FG, 1:4000); anti-H + -ATPase antibody (PHYTOAB, PHY2285A, 1:2000); anti-Rbcl antibody (Agrisera, AS03037, 1:4000); anti-H3 antibody (Abmart, P30266M, 1:3000); anti-β-Tubulin antibody (Abmart, M30109M, 1:4000); anti-GFP antibody (Abmart, M20004, 1:3000); anti-PIF4 antibody (Abiocode, R2534-4, 1:2000); anti-GST antibody (Abmart, M20007, 1:4000); anti-His antibody (GNI, GNI4110-HS, 1:4000); Alexa Fluor 488-labeled Goat Anti-Mouse IgG antibodies (Beyotime, A0428, 1:200); Glutathione Resin (GenScript, L00206); anti-GFP Affinity beads (Smart-Lifesciences, SA070001); rProtein A Beads 4FF (Smart-Lifesciences, SA015005); Ni NTA Beads 6FF (Smart-Lifesciences, SA005025).

## Reporting summary

Further information on research design is available in the Nature Portfolio Reporting Summary linked to this article.

## Data availability

The raw and processed RNA-seq data generated in this study have been deposited in the NCBI GEO database (https://www.ncbi.nlm.nih.gov/gds) with accession number GSE226205. Source data are provided with this paper.

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

## Acknowledgements

We thank Dr. Nam-Hai Chua, Dr. Rajeev J. Ram, and Dr. In-Cheol Jang (Singapore-MIT Alliance for Research and Technology) for the generous gifts of *phyB-9^BC*. We are grateful to the AraShare for the T-DNA insertion lines used in these studies. We thank Biorun Biogical Company for providing CRISPR/Cas9 technology. This research was supported by the National Natural Science Foundation of China (NSFC32030018) to L.L.

## Author contributions

L.L. designed the experiments; S.H. and Y.M. performed most of the experiments with the assistance from Y. Xu, Y. Xie, P.L. and J.Y.; S.H., Y.M., and L.L. analyzed the data; J.Y. helped with the figures; J.G. participated in discussions; L.L. wrote the manuscript.

## Competing interests

The authors declare no competing interests.
