## [Peer Review File · Nature Communications]

REVIEWER COMMENTS

Reviewer #1 (Remarks to the Author):

The authors analyzed some of enigmatic RTFL/DVL peptide family and in particular they studied OX and loss-of-function(l.o.f) mutant of RTFL18 and RTFL21. Because of high redundancy of roles of more than 20 members of RTFL/DVL peptides in Arabidopsis genome, to date, we knew phenotypes of over-expressors. Here they revealed that shade avoidance syndrome is somehow accelerated in the *rtfl18* and *rtfl21* l.o.f. mutants. This finding in combination with the finding of shade-induced nature of RTFL18/21 mRNAs is interesting.

But I have several concerns on this manuscript as listed below:

1. Fig.2a This is too small and I cannot recognize difference in the expression levels between leaf lamina and leaf petiole. Please magnify one of representative leaf.
2. Related to above, could not authors observe SAM regions in more magnified view? Because RTFL/DVL peptides are known to decrease cell division in leaf lamina if expressed, presence/absence of the RTFL18/21 expression in SAM and leaf primordia (not matured leaves) is a key to understand the roles of RTFL18/21.
3. 35::Flag-RTFL18 transgenic lines (line 136). Because RTFL peptides are small, it is plausible that tag fusion affects the function. Did authors observe any phenotype in this transgenic line? I am curious why the authors did not mention to it.
4. Fig. 3. This data is key, but I felt that shade avoidance syndrome (SAS) occurred even in RTFL18 ox lines. The shorter lengths under the shade might be due to short petiole phenotype, therefore the authors are requested to show the ratio of after and before shade treatment, in addition to the absolute length data here. Was elongation ratio decreased in *ox*? Because RTFL-*ox* suppresses cell division, we must strictly distinguish the cell proliferation-based phenotype and cell elongation-based phenotype (only the latter is the SAS).
5. Fig. 3d. Which domain was disrupted for RTFL18 and 21? In the so-called RTFL domain?
6. Fig. 4. YFP protein size is large in comparison with the small RTFL peptide and it might be plausible that YFP fusion warped interaction between RTFL18 and BSKs. Are authors confident on this interaction data? Again did authors confirmed the over-expression phenotype in Arabidopsis using this YFP-fused version?
7. Similar to the above #4, data in Fig. 5g is difficult to interpret. As the authors wrote the data on *rtfl18/rtfl21* was only "slightly enhanced" (Line 235). Again the authors are requested to show the response as ratio, too.
8. Discussion. Based on the data, the regulation of SAS by RTFL18/21 seems to be curious, namely, shade signal induces RTFL18/21 that suppress SAS. On this point the authors wrote "the findings indicate a negative feedback mechanism mediated by PIFs with RTFL18 to prevent excessive elongation of petioles under shade conditions" (Line 343). But (1) the genetic cascade shown in this manuscript is NOT 'feedback' loop structure; (2) if induction of RTFL18 by shade is rapid, elongation of petioles could not start. For example if the induction of RTFL18 is slow and there are some time lag between the triggering SAS and expression of RTFL18, some fine tuning role could be expected. But are there any evidence on such time lag?

Minor comments:

1. Italicized Arabidopsis means in a strict sense the genus Arabidopsis but not *A. thaliana*. Therefore I recommend the authors to use roman-styled Arabidopsis or italicized *A. thaliana*.
2. The latin name *A. tumefaciens* was recently changed into *Rhizobium radiobacter*.
3. '*' in *dvl1-1D*BSK3ox*, *rtfl18*rtfl21* and so on. Does this symbol mean double homozygous genotype or just a F1 cross?
4. Line 231. 'RTFL18 genetic materials'.... What?
5. Line 245. "endogenous antibody against PIF4". What is the 'endogenous'? No description in the Materials and Methods.
6. Line 303 "in BR levels". I think that there is no evidence on the regulation of levels of BR itself here.
7. Line 318 "on the basis of different shade light intensities". Again I think that there is no evidence on this idea.
8. Celsius degree symbol is wrongly typed.

Reviewer #2 (Remarks to the Author):

Huang et al present a very noteworthy paper on the finetuning of shade-avoidance responses in plants. The work answers long-standing questions on the mechanism-of-action of the RTFL peptides, and places the function of these peptides within a physiological context. I predict that the manuscript has the potential to become highly cited. The work is generally of very high quality and I only have one major reservation.

In Figure 4 F-I, the authors discuss a predicted structure of BSK3 with RTFL18. Personally, I think this data is quite weak. Firstly, it is not clear how these predictions were made. Secondly, the authors go into a lot of detail, highlighting several key residues for the interaction in their predicted structure, but none of these have been experimentally validated. I don't think the model is strictly needed for the manuscript. If the authors would like to make claims on which residues are responsible for binding they should perform extra experiments to confirm this.

Besides this point, I only had comments that I feel could improve the clarity, interpretation or reproducibility of these results:

As different kinds of shade (supplementary far-red vs reduced red / blue) trigger different molecular responses, it would be useful if the authors could be more specific about the type of shade cue given to plants within each figure.

41. "reduced leaf development" Could the authors be more specific?

Figure 1 b /c. It could be made clearer that the data in these figures is the same, just normalised differently. T-tests are not suitable for multiple comparisons, and so the authors should use a statistical test that does account for this (as they do for other gene expression data). Please also indicate if the data are technical (pipetting) or biological repeats to allow the reader to determine the source of variation.

Figure 1h. The genotypes used for the ChIP (35S:PIF4/7-Flash) could be mentioned here.

Figure 4a. This image is missing a scale bar.

Figure 4e. Maybe mention that this is SUMO-HIS-RFTL18

It may be due to the poor resolution in the images in the pdf, but to me it looks like pRTL18:LUC is highly abundant in the hypocotyl and roots of +FR-treated plants (Fig2.a.II). If this is actually signal from the WL image, the authors could consider splitting these channels.

Line 228: "We found that overexpression of RTFL18 did not affect the expression levels of BSKs (Supplementary Fig. 5f) and did not promote the degradation of the BSK3/6 protein (Supplementary Fig. 5g)". I feel that the conclusion about degradation is a little strong, given that the comparison is being made between two single independent transformants.

Line 248: "In addition, the level of PIF4 protein was lower in the bsk3 and bsk6 mutants and higher in the BSK3- and BSK6-overexpressing lines". Only one of the supplied blots shows reduced PIF4 abundance in the bsk6 mutant. I would like to see this conclusion softened accordingly.

Line 250: rescued the reduction in PIF4 protein levels in the dvl1-1D line?

Line 260: Fig S.6e is data on hypocotyl length but this is not discussed in the MS.

Line 315: Among these BSKs, only the expression of BSK5 was induced by shade and by PIFs. Which paper are the authors referring to here?

The formatting of the supplementary table should be re-assessed.

Reviewer #3 (Remarks to the Author):

In this manuscript ROT FOUR LIKE (RTFL) peptides are proposed to regulate shade avoidance response to supplemental far-red light. It is shown that some of the RTFLs are transcriptionally induced in a PIF-dependent manner, and their peptide products interact with the brassinosteroid signaling pathway, which is required for shade avoidance. The authors propose a novel pathway of RTFL18-BSK3/6-PIF4. This is an interesting and substantial body of work, but there are also many shortcomings, big and small, that need to be considered.

Major comments:

1. The physiological importance of cotyledon petiole elongation is not made clear in this manuscript. Why would this be a physiologically relevant read-out, is there any functional role for this tissue in a plant's life? In the manuscript, you use cotyledon petiole length and petiole length interchangeably. However, considering the developmental differences between the two organs, the distinction between the two needs to be clear every time you refer to it in the text e.g. line 234 and 236.
2. In the abstract already you claim the "RTFL18-BSK3/6-PIF4 module as a novel signaling cascade that prevents excessive activation of the shade avoidance response".
 - a. The RTFL18 part of this 'novel cascade' is not proven to be causal (rtfl18 mutants have no phenotype for shade avoidance whatsoever, fig.3), so this should be about RTFL21. However, for RTFL21 the link with BSK3/6 is not supported by data. In other words, this claim cannot be made from your data.
 - b. With respect to the rest of the cascade (BSK – PIF), this has been shown previously for a related BSK (BSK5). I do not consider the published BSK5 work to take away novelty from your work, but it is important to acknowledge it.
3. Your protein data, of which there are many in this manuscript, lack reproduction. It is important to have sufficient biological replicates, each of them quantified, and then show the average relative protein abundance, including SD, essentially just like you do for phenotypes.
4. Following up on the Westerns: it needs to be described what the numbers underneath each blot scan identify. I am assuming they are a quantification of that very blot of some sort, but it is not written. Do you show mean pixel intensity / total pixel intensity / band size?
5. You state (already in the Abstract) that RTFLs repress activity of BSKs and PIF4. You have not shown activities of either of these groups of proteins, and this is therefore a major over-interpretation of your data. This should be fixed in your writing, or by including data that prove the point.
6. In this manuscript, the word shade is used, while you use a supplemental Far-Red treatment in all of the experiments. This is not a shade treatment, and should therefore not be identified as such in this manuscript. The used treatment would be considered mimicking neighbor proximity, but it would be fine to refer to it as a low R:FR treatment or FR-enrichment. Calling it a shade treatment is incorrect, even though others have done so before.
7. Following on the above, details about your light treatments need to be provided: lamp types (brand, etc) and full spectral composition of your light treatments (in $\mu\text{mol m}^{-2} \text{s}^{-1} \text{nm}^{-1}$) must be provided.
8. For all qRT-PCR experiments and ChIP-qPCR experiments, whole seedlings were used, except for figure 2b. Considering the focus on the young leaf petioles and the cotyledon petioles, these experiments would need to be performed ideally on these petiole tissues, or minimally on shoot tissue only. I realize this is a substantial new effort, but right now the data can be confounded in many ways. Additionally, it is not clear from the materials and methods if the protein work in figures 2c, 6a-e and supplemental figure 6g was performed on whole seedlings or shoot, as there is no description of these experiments in the methods section. This has to be fixed in the methods section, and should also be clear from the caption.
9. There are several instances where you overstep your data, some are listed below but the list is

probably not complete:

- a. Line 140-143: Based on the images, it is difficult to assess the subcellular localization of both the 35S::GFP-RTFL18 and FM4-64. In the merged image, both green and red are visible and they barely overlap (despite your statement that they do). Moreover, this experiment was done by transiently expressing both the control 35S::GFP and the 35S::GFP-RTFL18 in *Nicotiana*, and imaged in the epidermal cells. In both instances your protein seems to localize (close to) the plasma membrane, perhaps in part because the cytoplasm is pushed against the membrane by a large vacuole. Taken together, the data are currently not sufficient to draw conclusions about a putative plasma membrane-localization of RTFL18.
- b. Line 162-167: You conclude that only the *rtfl18 * rtfl21* mutant showed a longer petiole length than WT, but without the data of single mutant *rtfl21* this conclusion is not warranted. Better to also include experimental data of single mutant *rtfl21*, just like you did for *rtfl18*.
- c. Section starting from Line 169: Although interaction is shown via luciferase complementary imaging, BiFC and semi in-vivo and in-vitro pulldown assays, there was no actual in-vivo experimental work done, nor is there an experiment done where the interaction is lost through manipulation. The in silico work gives a lot of potential starting points to perform such experiments, yet the mechanistic, experimental proof is lacking.
- d. Line 206 "RTFL21 also interacts with BSK3/6": There is no experimental data to substantiate this statement. Rephrase to "Structural superposition predicts that RTFL21 could interact with BSK3/6".
- e. Line 209-211: These are all very speculative in nature, and are not supported by any data that is not in silico. If you really want to make these statement, they would be better suited for the Discussion.

10. The experiments done in 5g and h are not described in the material and methods section. Secondly, data are missing on the leaf petiole (only those for the cotyledon are presented).

11. In your Quantitative RT-PCR experiments you use only one reference gene. This is not super-robust and makes you very dependent on its stability throughout your treatments, tissues and genotypes. Have you verified this?

12. You draw conclusions about PIF4 action in lines 258-259 based on the expression of one PIF4 target gene, PRE1. This statement should either be removed, or experiments should be conducted that allow such a strong statement. PRE1 is involved in many routes and not just regulated by PIF4, so more evidence would be needed than just the expression of this individual gene.

13. A schematic (flow chart) summarizing the results and the proposed pathway would be of great help to the reader.

Minor comments:

1. The sentence structure interferes with comprehensibility at times, see e.g.
 - a. Line 34, do you mean energy source? Or energy as a signal in itself?
 - b. Lines 36-40
 - c. At line 49, do you mean: "... activation downstream of phyB", instead of ".. activation of downstream phyB."?
 - d. Line 234: "similar to their response to *bsk3* and *br1-301*"
2. Line 111 refers to five RTFLs, without looking at the figure it is unclear about which five RTFLs you are mentioning.
3. You present data from previous work in figures 1a, supplemental figure 1b, and some other supplemental figures without a need for it. I suggest to restrict to quantitative RT-PCR data that you focus on mostly anyway.
4. In figure 1c,d, and e, the Y-axis mentions a ratio, however, both components of this ratio are plotted in the figure. This is impossible, the ratio per definition excludes the two bars.
5. In various figures you have provided phenotypic quantifications alongside with photographic images. This is a good idea, but it has become confusing: you are probably using the following order of genotypes as in the graph underneath, but in fact a direct comparison is impossible: in the graph you have WL and +FR directly coupled per genotype, whereas in the photos you first show all the genotypes in WL and then show all the genotypes in +FR. This is very confusing in Figure 3. I suggest you do it as you did in Fig.5a,b,d,e, but then include genotype names underneath your photos so as to be make it entirely clear what is what.

6. It would help the reader to understand the line of reasoning if some choices are better supported by experimental data. For instance the selection of BKS3 is not explained, as it is missing from supplemental figure 4b.
7. In supplementary figure 5g, it is not clear if the bar graphs are quantifications of the western blots on the right, or if they are independent qRT-PCRs.

Authors' Responses to Reviewers' Comments (NCOMMS-23-03409)

Reviewer #1:

The authors analyzed some of enigmatic RTFL/DVL peptide family and in particular they studied OX and loss-of-function(l.o.f) mutant of RTFL18 and RTFL21. Because of high redundancy of roles of more than 20 members of RTFL/DVL peptides in Arabidopsis genome, to date, we knew phenotypes of over-expressors. Here they revealed that shade avoidance syndrome is somehow accelerated in the *rtfl18* and *rtfl21* l.o.f. mutants. This finding in combination with the finding of shade-induced nature of RTFL18/21 mRNAs is interesting.

Response: We thank Reviewer #1 for carefully considering this work. We have addressed his or her specific remarks and questions in our revised manuscript, as outlined below.

But I have several concerns on this manuscript as listed below:

1. Fig.2a This is too small and I cannot recognize difference in the expression levels between leaf lamina and leaf petiole. Please magnify one of representative leaf.

Response: Please check revised Fig. 2a I, II, III and IV. The expressions and low R:FR induction of *RTFL18* can be observed in cotyledon, cotyledon petiole, leaf and leaf petiole.

2. Related to above, could not authors observe SAM regions in more magnified view? Because RTFL/DVL peptides are known to decrease cell division in leaf lamina if expressed, presence/absence of the RTFL18/21 expression in SAM and leaf primordia (not matured leaves) is a key to understand the roles of RTFL18/21.

Response: As the reviewer suggested, we observed the leaf primordia and SAM of 5-day-old *dv1-1D* and *rtfl18*rtfl21* double-mutant seedlings (Fig. R1). We found reduced leaf primordium length in *dv1-1D*, which is consistent with the leaf phenotype in a previous report¹, and a smaller SAM in *dv1-1D*.

Fig. R1 | Leaf primordium and shoot apical meristem (SAM) of *dv11-1D* and *rtf18*rtf21* under white light (WL) and low R:FR conditions. a, Leaf primordium of Col-0, *dv11-1D* and *rtf18*rtf21* under white light (WL) and low R:FR conditions. The left panels show the leaf primordium. The scale bar represents 1 mm. The right panel displays corresponding leaf primordium lengths ($n > 16$). Seedlings grown for 4 days under white light were transferred to low R:FR conditions or continued to be grown under white light for 1 day. The error bars indicate the SEM. The letters indicate significant differences between mean values (two-way ANOVA: Tukey's multiple comparisons test, $P < 0.01$), and groups with the same letters are not significantly different. b, SAM of Col-0, *dv11-1D* and *rtf18*rtf21* under WL and low R:FR conditions. The left panels show SAM images. The scale bar represents 50 μm . The right panel displays the corresponding cell number of SAM ($n > 6$). Seedlings grown for 4 days under white light were transferred to low R:FR conditions or remained under white light for 1 day. The error bars indicate the SEM. The letters indicate significant differences between mean values (two-way ANOVA: Tukey's multiple comparisons test, $P < 0.01$), and groups with the same letters are not significantly different.

3. 35::Flag-RTFL18 transgenic lines (line 136). Because RTFL peptides are small, it is plausible that tag fusion affects the function. Did authors observe any phenotype in this transgenic line? I am curious why the authors did not mention to it.

Response: To determine whether Flag tag fusion affects the function of RTFL18, we observed the phenotypes (cotyledon petiole length, leaf petiole length and hypocotyl length) of *35S::Flag-RTFL18* and *35S::RTFL18-Flag* transgenic lines. These Flag tag fused transgenic lines displayed similar phenotypes to *35S::RTFL18* transgenic lines and *dvl1-1D* (a T-DNA insertion in the promoter of RTFL18 as an overexpression allele) (revised Fig. 3a, b, c and Supplementary Fig. 3c), indicating that the Flag tag does not affect the function of RTFL18 in the phenotypes we detected.

4. Fig. 3. This data is key, but I felt that shade avoidance syndrome (SAS) occurred even in RTFL18 ox lines. The shorter lengths under the shade might be due to short petiole phenotype, therefore the authors are requested to show the ratio of after and before shade treatment, in addition to the absolute length data here. Was elongation ratio decreased in ox?

Response: To clearly show the low R:FR response, we calculated the ratio of cotyledon petiole length and petiole length under WL and low R:FR conditions. Based on the ratio, we found that the low R:FR response on cotyledon petiole length and petiole elongation was reduced in *RTFL18-overexpressing* plants (revised Fig. 3b, c).

Because RTFL-ox suppresses cell division, we must strictly distinguish the cell proliferation-based phenotype and cell elongation-based phenotype (only the latter is the SAS).

Response: As the reviewer suggested, we measured the cell length and cell

number in cotyledon petioles and leaf petioles (revised Supplementary Fig. 3e and Fig. R2). The results showed that low R:FR-induced cotyledon petiole and petiole cell elongations decreased in *dv1-1D*, and increased in *rtfl18 * rtfl21*, which supported that RTFL18 is involved in low R:FR-induced cotyledon petiole and leaf petiole elongation.

Fig. R2 | The cell number in cotyledon petioles and leaf petioles. a, The cell number in cotyledon petioles of Col-0, *dv1-1D*, and *rtfl18*rtfl21*. b, The cell number in leaf petioles of Col-0, *dv1-1D*, and *rtfl18*rtfl21*. The cell number in the central row of the petiole longitudinal axis was counted. Seedlings grown for 3 days under white light were transferred to low R:FR conditions or remained under white light for 8 days. The error bars indicate the SEM (n>16). Letters indicate significant differences between mean values (two-way ANOVA: Tukey's multiple comparisons test, $P < 0.01$), and groups with the same letters are not significantly different.

5. Fig. 3d. Which domain was disrupted for RTFL18 and 21? In the so-called RTFL domain?

Response: A one-nucleotide insertion in the exon of *RTFL18* in *rtfl18-1* and *rtfl18-2*, a 10-nucleotide deletion in *rtfl18-3*, and a one-nucleotide insertion in the exon of *RTFL21* in the *rtfl21* and *rtfl18-2*rtfl21* mutants, led to premature stop codons and generated truncated proteins (revised Fig. 3d). Consequently, these mutants lose the RTF functional area on the C-terminal. We have modified Fig. 3d.

6. Fig. 4. YFP protein size is large in comparison with the small RTFL peptide and it might be plausible that YFP fusion warped interaction between RTFL18 and BSKs. Are authors confident on this interaction data? Again did authors confirmed the over-expression phenotype in Arabidopsis using this YFP-fused version?

Response: First, the interactions between RTFL18 and BSKs were confirmed by a LCI assay, an *in vitro* pull down assay, a semi-*in vitro* pull down assay, and a newly added CoIP assay in tobacco and Arabidopsis, in addition to the BiFC assay (revised Fig. 4). In our BiFC assay, nYFP was fused to the N-terminus of RTFL18 and this fusion did not affect the interaction with BSKs. Second, we did not generate the YFP-RTFL18 transgenic lines. DVL1/RTFL18 with green fluorescent protein (GFP) fused to the N-terminus (GFP-DVL1) has been reported to be fully functional².

7. Similar to the above #4, data in Fig. 5g is difficult to interpret. As the authors wrote the data on *rtfl18/rtfl21* was only "slightly enhanced" (Line 235). Again the authors are requested to show the response as ratio, too.

Response: In Fig. 5g, the absolute length data are shown on the left side and the ratio of cotyledon petiole lengths between eBL treatments and without treatment is shown on the right side. Based on these results, we found that overexpression of *RTFL18* leads to reduced sensitivity to eBL, similar to *bsk3* and *bri1-301*, but no significant change was observed in the response of *rtfl18*rtfl21* to eBL compared to Col-0. The related sentences have been revised (Line 257).

8. Discussion. Based on the data, the regulation of SAS by RTFL18/21 seems to be curious, namely, shade signal induces RTFL18/21 that suppress SAS. On this point the authors wrote "the findings indicate a negative feedback mechanism mediated by PIFs with RTFL18 to prevent excessive elongation of petioles under shade conditions" (Line 343). But (1) the genetic cascade shown

in this manuscript is NOT 'feedback' loop structure;

Response: Thank you for pointing this out. In our current study, PIFs positively regulated the expression of *RTFL18*, and *RTFL18* negatively regulated the protein levels of PIF4 through interacting with BSK3/6. We checked the expression and protein levels of PIF4 in *PIF4ox* and *dvl1-1D*PIF4ox* (revised Supplementary Fig. 6f). We also rechecked the phenotype of *dvl1-1D*PIF4ox*, and found that overexpression of PIF4 partially rescued the shorter cotyledon petiole and petiole in *dvl1-1D* (revised Fig. 6g, h). We have modified this sentence as follows: “The findings indicate that a negative mechanism mediated by *RTFL18* occurs to prevent petioles from excessively elongating under low R:FR conditions”.

(2) if induction of *RTFL18* by shade is rapid, elongation of petioles could not start. For example if the induction of *RTFL18* is slow and there are some time lag between the triggering SAS and expression of *RTFL18*, some fine tuning role could be expected. But are there any evidence on such time lag?

Response: We agree that a time lag might occur from the induction of *RTFL18* to the effect on PIF4 protein level and to changes on phenotypes. However, it is difficult to determine an accurate time lag due to the different sensitivities of the assays we used for monitoring the induction of *RTFL18*, the stability of PIF4 protein, and changes on phenotype. We performed several experiments to answer this question. As shown in Fig. R3, the induction of *RTFL18* was detected after 1 h of low R:FR treatment by qRT-PCR and western blotting, and the negative effect of *RTFL18* on PIF4 protein levels was detected after 6 h of low R:FR treatment by western blotting which might result from the accumulation time of detectable PIF4 protein levels under low R:FR. We can monitor the growth rate of hypocotyl elongation by DynaPlant (not petiole elongation thus far), and we found that the effect of *RTFL18* appeared after 3 h of low R:FR treatment.

Fig. R3 | The expression of RTFL18/PIF4 and growth rate of hypocotyl elongation after low R:FR treatment. a, The transcriptional level of *RTFL18* after low R:FR treatment was measured by qRT-PCR. The expression levels of *RTFL18* were normalized against the expression of the reference *AT2G39960*. The error bars indicate the SD (n=3). b, The protein level of RTFL18 after low R:FR treatment was measured by western blotting. c, The protein level of PIF4 after low R:FR treatment was measured by western blotting. d, The growth rate of hypocotyl elongation after low R:FR treatment was measured by DynaPlant. e, The average growth rate of hypocotyl elongation after low R:FR treatment. In a, b, and c, seedlings grown for 6 days under white light were transferred to low R:FR conditions for different times. In d and e, seedlings grown for 4 days under white light were transferred to low R:FR conditions for different times. The error bars indicate the SEM (n > 16). The asterisks indicate significant differences to Col-0. (Student's *t* test, ****P* < 0.001, ns indicates no significance).

Minor comments:

1. Italicized Arabidopsis means in a strict sense the genus Arabidopsis but not *A. thaliana*. Therefore I recommend the authors to use roman-styled Arabidopsis or italicized *A. thaliana*.

Response: Thanks, corrected.

2. The latin name *A. tumefaciens* was recently changed into *Rhizobium radiobacter*.

Response: Done.

3. '*' in *dvl1-1D*BSK3ox*, *rtfl18*rtfl21* and so on. Does this symbol mean double homozygous genotype or just a F1 cross?

Response: *rtfl18*rtfl21* is homozygous. For *dvl1-1D*BSK3ox*, *dvl1-1D*BSK6ox* and *dvl1-1D*PIF4ox*, we selected *dvl1-1D* as homozygous lines by PCR genotyping and overexpressing BSK3/6 or PIF4 according to resistance to antibiotics and expression levels measured by qRT-PCR and western blotting. The related information has been added into Materials and Methods (Line 419).

4. Line 231. 'RTFL18 genetic materials".... What?

Response: We have listed the detailed genotypes "*dvl1-1D*, *35S:RTFL18* and *rtfl18*rtfl21*" in our revised manuscript.

5. Line 245. "endogenous antibody against PIF4". What is the 'endogenous'? No description in the Materials and Methods.

Response: The anti-PIF4 antibody we used is a commercial antibody from Abiocode (R2534-4). We have added the section "Antibody information" in the Materials and Methods.

6. Line 303 "in BR levels". I think that there is no evidence on the regulation of levels of BR itself here.

Response: Thanks. We have modified the related sentence in our revised manuscript.

7. Line 318 "on the basis of different shade light intensities". Again I think that there is no evidence on this idea.

Response: We agree with the reviewer and have removed this sentence in our revised manuscript.

8. Celsius degree symbol is wrongly typed.

Response: Thanks, corrected.

Reviewer #2 (Remarks to the Author):

Huang et al present a very noteworthy paper on the finetuning of shade-avoidance responses in plants. The work answers long-standing questions on the mechanism-of-action of the RTFL peptides, and places the function of these peptides within a physiological context. I predict that the manuscript has the potential to become highly cited.

Response: We thank Reviewer #2 for appreciations about this work. We have addressed his or her specific concerns in our revised manuscript, as outlined below.

The work is generally of very high quality and I only have one major reservation. In Figure 4 F-I, the authors discuss a predicted structure of BSK3 with RTFL18. Personally, I think this data is quite weak. Firstly, it is not clear how these predictions were made. Secondly, the authors go into a lot of detail, highlighting several key residues for the interaction in their predicted structure, but none of these have been experimentally validated. I don't think the model is strictly needed for the manuscript. If the authors would like to make claims on which residues are responsible for binding they should perform extra experiments to confirm this.

Response: Thanks. We have added a detailed description of the structure prediction as "Modelling of the RTFL/BSK complex" in the Methods. Furthermore, to verify the key residues that interact in our predicted structure, we detected the interactions between RTFL18 M1 (I39A, M43A, L47A) and RTFL18 M2 (I40A, H50A, D51A) with BSK3, and BSK3 M1 (E368A, K385A, R393A), BSK3 M2 (W374A, M378A, L382A), and BSK3 M3 (E368A, K385A, R393A, T374A, M378A, L382A) with RTFL18 by *in vitro* pull down assays. As shown in revised Fig. 4l and 4m, the faint bands indicated that mutated RTFL18 exhibits reduced affinity with BSK3 or mutated BSK3 with RTFL18. These additional data demonstrated the importance of these key residues in the interactions.

Besides this point, I only had comments that I feel could improve the clarity, interpretation or reproducibility of these results:

As different kinds of shade (supplementary far-red vs reduced red / blue) trigger different molecular responses, it would be useful if the authors could be more specific about the type of shade cue given to plants within each figure.

Response: Thanks. The low R:FR condition we used was listed in the methods, and we also mentioned it within each revised figure legend.

41. “reduced leaf development” Could the authors be more specific?

Response: We have modified “a rapid arrest in leaf development” according to the reference³ in our revised manuscript.

Figure 1 b /c. It could be made clearer that the data in these figures is the same, just normalised differently. T-tests are not suitable for multiple comparisons, and so the authors should use a statistical test that does account for this (as they do for other gene expression data). Please also indicate if the data are technical (pipetting) or biological repeats to allow the reader to determine the source of variation.

Response: As the reviewer suggested, we have modified the statistical tests (one-way ANOVA) in Fig. 1b. The data represent biological repeats. This information has been added to the figure legends.

Figure 1h. The genotypes used for the ChIP (35S:PIF4/7-Flash) could be mentioned here.

Response: The *35S:PIF4/7-Flash* transgenic lines have been mentioned in Fig. 1h legends and the methods section (line 450).

Figure 4a. This image is missing a scale bar.

Response: Thanks, we have added the scale bar in Fig. 4a.

Figure 4e. Maybe mention that this is SUMO-HIS-RFTL18

Response: Thanks, we have added this description in revised figure legend.

It may be due to the poor resolution in the images in the pdf, but to me it looks

like pRTL18:LUC is highly abundant in the hypocotyl and roots of +FR-treated plants (Fig2.a.II). If this is actually signal from the WL image, the authors could consider splitting these channels.

Response: Thanks. We have selected more representative plants and uploaded the images with higher resolution in revised Fig. 2a.

Line 228: “We found that overexpression of RTFL18 did not affect the expression levels of BSKs (Supplementary Fig. 5f) and did not promote the degradation of the BSK3/6 protein (Supplementary Fig. 5g)”. I feel that the conclusion about degradation is a little strong, given that the comparison is being made between two single independent transformants.

Response: We agree with the reviewer. To confirm the effect of RTFL18 on the protein level of BSKs, we screened *BSK3ox* and *dv1-1D * BSK3ox* with similar expression levels of *BSK3*, *BSK6ox* and *dv1-1D * BSK6ox* with similar expression levels of *BSK6* (measured by qRT-PCR). Considering the negative effect of RTFL18 on BR signaling, we expected that RTFL18 would exert a negative effect. However, we found that overexpression of *RTFL18* did not cause a negative effect on the protein level of BSK3/6 in two pairs of independent lines (revised Supplementary Fig.5g, h). We concluded that RTFL18 negatively regulates BR signaling through the interaction between BSKs, not BSKs themselves.

Line 248: “In addition, the level of PIF4 protein was lower in the *bsk3* and *bsk6* mutants and higher in the *BSK3*- and *BSK6*-overexpressing lines”. Only one of the supplied blots shows reduced PIF4 abundance in the *bsk6* mutant. I would like to see this conclusion softened accordingly.

Response: To strengthen our conclusion, we have provided three biological replicates and quantified them in revised Fig. 6a-e and Supplementary Fig. 6a-e.

Line 250: rescued the reduction in PIF4 protein levels in the *dv1-1D* line?

Response: Per the reviewer’s suggestion, we have modified the description in the text as “*BSK3/6* overexpression rescued the PIF4 protein level in the *dv1-*

1D background”.

Line 260: Fig S.6e is data on hypocotyl length but this is not discussed in the MS.

Response: We have mentioned the hypocotyl phenotype of Supplementary Fig. 6f in the revised MS.

Line 315: Among these BSKs, only the expression of BSK5 was induced by shade and by PIFs. Which paper are the authors referring to here?

Response: We have added reference⁴ in line 358.

The formatting of the supplementary table should be re-assessed.

Response: We have modified the format of the supplementary table.

Reviewer #3

In this manuscript ROT FOUR LIKE (RTFL) peptides are proposed to regulate shade avoidance response to supplemental far-red light. It is shown that some of the RTFLs are transcriptionally induced in a PIF-dependent manner, and their peptide products interact with the brassinosteroid signaling pathway, which is required for shade avoidance. The authors propose a novel pathway of RTFL18-BSK3/6-PIF4. This is an interesting and substantial body of work, but there are also many shortcomings, big and small, that need to be considered.

Response: We thank Reviewer #3 for careful reading and helpful comments on our manuscript. We have further addressed his or her new concerns in our revised manuscript and outlined our responses as below.

Major comments:

1. The physiological importance of cotyledon petiole elongation is not made clear in this manuscript. Why would this be a physiological relevant read-out, is there any functional role for this tissue in a plants life? In the manuscript, you use cotyledon petiole length and petiole length interchangeably. However, considering the developmental differences between the two organs, the distinction between the two needs to be clear every time you refer to it in the

text e.g. line 234 and 236.

Response: The length of cotyledon petioles has been detected in various studies in different situations, such as during plant development processes⁵, response to temperature⁶, and response to light and hormones^{7,8}. In the current study, low R:FR light induced the elongation of both cotyledonary petioles and petioles. We have not found solid evidence to support the physiological relevance between these two readouts thus far. We displayed both the length of the petiole and the petiole in the most phenotype-related experiments, except the eBL and bikinin response experiments. As shown in Fig. 5g and 5h, the seedlings cannot survive long-term eBL and bikinin treatments, which are necessary for measuring the response of petioles. Therefore, we only showed the cotyledon petiole length following the protocol described before^{7,9}. We have carefully checked and corrected the description of the cotyledon petiole and petiole in the revised manuscript.

2. In the abstract already you claim the “RTFL18-BSK3/6-PIF4 module as a novel signaling cascade that prevents excessive activation of the shade avoidance response”.

a. The RTFL18 part of this ‘novel cascade’ is not proven to be causal (*rtfl18* mutants have no phenotype for shade avoidance whatsoever, fig.3), so this should be about RTFL21. However, for RTFL21 the link with BSK3/6 is not supported by data. In other words, this claim cannot be made from your data.

Response: It has been reported that RTFL family peptides have functional redundancy¹. We obtained a *rtfl21* single mutant and found that the single mutant did not display any significant difference from Col-0 (Fig. 3f, g, Supplementary Fig. 3d). Moreover, RTFL21 (*DVL3*) RNAi plants have been reported to exhibit no phenotype of functional loss¹. The *rtfl4-1*, *osrtfl1-1*, and *osrtfl2-1* mutants did not exhibit significant phenotypes¹⁰. The plants overexpressing *RTFL19* (*DVL2*), *RTFL21* (*DVL3*), *RTFL17* (*DVL4*), and *RTFL15* (*DVL5*) exhibited similar phenotypes, such as reduced and rounded

rosette leaves, clustered inflorescences, and forked fruits¹. Based on our structure predictions, the key residues are highly conserved among RTFL family members that mediate the interactions with BSKs. Therefore, we believe that the petiole phenotype of the *rtfl18*rtfl21* double mutant is caused by two RTFL mutations.

b. With respect to the rest of the cascade (BSK – PIF), this has been shown previously for a related BSK (BSK5). I do not consider the published BSK5 work to take away novelty from your work, but it is important to acknowledge it.

Response: Yes, we agree with the reviewer. The published BSK5-PIF work is commendable and acknowledgeable. We have described the work in the introduction and discussion (Line 63, 357).

3. Your protein data, of which there are many in this manuscript, lack reproduction. It is important to have sufficient biological replicates, each of them quantified, and then show the average relative protein abundance, including SD, essentially just like you do for phenotypes.

Response: We have added more biological replicates of the protein results and quantified them in revised Fig. 6a-e and Supplementary Fig. 6a-e.

4. Following up on the Westerns: it needs to be described what the numbers underneath each blot scan identify. I am assuming they are a quantification of that very blot of some sort, but it is not written. Do you show mean pixel intensity / total pixel intensity / band size?

Response: The numbers underneath each blot scan represent the ratio of the target protein to the internal reference protein, which are the relative mean gray values. We have added a “Western blotting and analysis” section in the materials and methods.

5. You state (already in the Abstract) that RTFLs repress activity of BSKs and PIF4. You have not shown activities of either of these groups of proteins, and

this is therefore a major over-interpretation of your data. This should be fixed in your writing, or by including data that proof the point.

Response: Thanks. We agree with the reviewer that we have not shown the activities of BSKs and PIF4. We have changed the inappropriate description as follows: RTFL peptides negatively regulate the SAS by interacting with BRASSINOSTEROID SIGNALING KINASEs (BSKs) and reducing the protein level of PHYTOCHROME INTERACTING FACTOR 4 (PIF4) in Arabidopsis.

6. In this manuscript, the word shade is used, while you use a supplemental Far-Red treatment in all of the experiments. This is not a shade treatment, and should therefore not be identified as such in this manuscript. The used treatment would be considered mimicking neighbor proximity, but it would be fine to refer to it as a low R:FR treatment or FR-enrichment. Calling it a shade treatment is incorrect, even though others have done so before.

Response: As the reviewer suggested, we have replaced shade treatment with low R: FR treatment.

7. Following on the above, details about your light treatments need to be provided: lamp types (brand, etc) and full spectral composition of your light treatments (in $\mu\text{mol m}^{-2} \text{s}^{-1} \text{nm}^{-1}$) must be provided.

Response: We described the lamp types in the Materials and Methods. We added the full spectral composition of your light treatments in the revised Supplementary Fig. 1b.

8. For all qRT-PCR experiments and ChiP-qPCR experiments, whole seedlings were used, except for figure 2b. Considering the focus on the young leaf petioles and the cotyledon petioles, these experiments would need to be performed ideally on these petiole tissues, or minimally on shoot tissue only. I realize this is a substantial new effort, but right now the data can be confounded in many ways.

Response: When we first focused on the low R:FR inductions of *RTFLs*, the dramatic responses on the expression of *RTFLs* stand out from several published RNA-seq studies using whole seedlings. We verified the expression and low R:FR induction of *RTFL18* in separated hypocotyls, cotyledons and petioles, as well as leaves and petioles (revised Fig.2b). We agree that if we use petiole-specific tissue, we might obtain a higher expression level of *RTFL18*, but it might not enhance low R:FR-inductions and the enrichment of PIFs on the promoter region of *RTFLs*. Low R:FR induction of *RTFL18* can also be detected in hypocotyls, but we did not observe a significantly longer hypocotyl phenotype in *rtfl18*rtfl21*. This might be due to functional redundancy among *RTFL* family members and tissue-specific expression patterns (Discussion, line 327). Additionally, in Fig. 6a-e and newly added Fig. 7, we used whole seedlings to detect the effect of *RTFL18* on the protein level of PIF4 and downstream gene expression under low R:FR. We observed corresponding changes after low R:FR treatment in the whole seedling.

Additionally, it is not clear from the materials and methods if the protein work in figures 2c, 6a-e and supplemental figure 6g was performed on whole seedlings or shoot, as there is no description of these experiments in the methods section. This has to be fixed in the methods section, and should also be clear from the caption.

Response: We added descriptions of these experiments in the “Western blotting and analysis” section in the methods and in the figure legends.

9. There are several instances where you overstep your data, some are listed below but the list is probably not complete:

a. Line 140-143: Based on the images, it is difficult to assess the subcellular localization of both the 35S::GFP-*RTFL18* and FM4-64. In the merged image, both green and red are visible and they barely overlap (despite your statement that they do). Moreover, this experiment was done by transiently expressing both the control 35S::GFP and the 35S::GFP-*RTFL18* in *Nicotiana*, and imaged

in the epidermal cells. In both instances your protein seems to localize (close to) the plasma membrane, perhaps in part because the cytoplasm is pushed against the membrane by a large vacuole. Taken together, the data are currently not sufficient to draw conclusions about a putative plasma membrane-localization of RTFL18.

Response: To further investigate the subcellular location of RTFL18, we (1) performed immunofluorescence and found that Flag-RTFL18 was localized on the cell membrane in *35s::Flag-RTFL18* transgenic lines (revised Fig. 2e); and (2) separated the cell membrane from other components and found that Flag-RTFL18 exists in the cell membrane and is not present in other cell components (revised Fig. 2f). Additionally, the RTFL family member ROT4 has been reported to be localized on the cell membrane¹¹. In summary, we believe that RTFL18 is localized on the cell membrane.

b. Line 162-167: You conclude that only the *rtfl18 * rtfl21* mutant showed a longer petiole length than WT, but without the data of single mutant *rtfl21* this conclusion is not warranted. Better to also include experimental data of single mutant *rtfl21*, just like you did for *rtfl18*.

Response: We obtained the *rtfl21* single mutant and did not find a significant defective phenotype (revised Fig. 3e, f, g, and Supplementary Fig. 3d).

c. Section starting from Line 169: Although interaction is shown via luciferase complementary imaging, BiFC and semi in-vivo and in-vitro pulldown assays, there was no actual in-vivo experimental work done, nor is there an experiment done where the interaction is lost through manipulation. The in silico work gives a lot of potential starting points to perform such experiments, yet the mechanistic, experimental proof is lacking.

Response: We added Co-IP results to solidify the interaction between RTFL18 and BSK3/6. In *N. benthamiana* leaves, Flag-RTFL18 proteins were immunoprecipitated with BSK3-GFP or BSK6-GFP, but not GFP only (Fig. 4f). In double overexpression Arabidopsis *BSK3-GFP * Flag-RTFL18* and *BSK6-GFP * Flag-RTFL18*, co-immunoprecipitation between BSK3/6-GFP and Flag-

RTFL18 also can be detected (Fig.4g).

We also mutated the key amino acid sites of the simulated interaction between BSK3 and RTFL18, and validated their interaction using an *in vitro* pull-down assay (Fig. 4l, m). The results showed that the interactions between amino acid mutated RTFL18 and BSK3 were decreased, and the interactions between amino acid mutated BSK3 and RTFL18 were also affected.

d. Line 206 “RTFL21 also interacts with BSK3/6”: There is no experimental data to substantiate this statement. Rephrase to “Structural superposition predicts that RTFL21 could interact with BSK3/6”.

Response: Yes, we agree with the reviewer and have changed the description to “RTFL21 might also interact with BSK3/6”.

e. Line 209-211: These are all very speculative in nature, and are not supported by any data that is not *in silico*. If you really want to make these statement, they would be better suited for the Discussion.

Response: To further investigate the importance of the key amino acid sites of the simulated interaction between BSK3 and RTFL18, we tested the interactions between mutated RTFL18 with BSK3, and mutated BSK3 with RTFL18 by *in vitro* pull-down assays (revised Fig. 4l, m).

10. The experiments done in 5g and h are not described in the material and methods section. Secondly, data are missing on the leaf petiole (only those for the cotyledon are presented).

Response: We added a “Sensitivity to eBL and bikinin” section related to Fig. 5g and 5h in the Materials and Methods. The seedlings cannot survive long-term eBL and bikinin treatments, which are necessary for measuring the response of petioles. Therefore, we only showed the cotyledon petiole length following the protocol described before⁷.

11. In your Quantitative RT-PCR experiments you use only one reference gene. This is not super-robust and makes you very dependent on its stability

throughout your treatments, tissues and genotypes. Have you verified this?

Response: We searched multiple published RNA-seq datasets and our RNA-seq data¹²⁻¹⁴ and found that the reference gene *AT2G39960* was stable after low R:FR treatment and in *dvl1-1D*, *pif4pif7*, and *bsk3bsk6* mutants (Fig. R4).

Fig. R4 Transcription level of *AT2G39960*. The error bars indicate the SD. Letters indicate significant differences between mean values (a, b, d, e, and f two-way ANOVA: Tukey's multiple comparisons test, $P < 0.01$; c one-way ANOVA, $P < 0.01$), and groups with the same letters are not significantly different.

12. You draw conclusions about PIF4 action in lines 258-259 based on the expression of one PIF4 target gene, *PRE1*. This statement should either be removed, or experiments should be conducted that allow such a strong statement. *PRE1* is involved in many routes and not just regulated by PIF4, so more evidence would be needed than just the expression of this individual gene.

Response: To investigate the effects of RTFL18 and BSK3/6 on low R:FR-responsive transcription, we conducted RNA-seq analysis in *dvl1-1D*, *bsk3bsk6* and *pif4pif7* (Fig. 7). We identified 370 low R:FR-responsive genes that were coregulated by RTFL18, BSK3/6, and PIF4/7 (Fig. 7c).

The secure token has been created to allow review of record GSE226205 while it remains in private status: sjppyuykhjsvnm.

13. A schematic (flow chart) summarizing the results and the proposed pathway would be of great help to the reader.

Response: We added a schematic in Supplementary Fig. 7e.

Minor comments:

1. The sentence structure interferes with comprehensibility at times, see e.g.

a. Line 34, do you mean energy source? Or energy as a signal in itself?

Response: This sentence has been changed to “Light provides energy for photosynthesis and also acts as an important environmental signal to affect growth and development throughout the whole life cycle of plants”.

b. Lines 36-40

Response: This sentence has been shortened.

c. At line 49, do you mean: “.... activation downstream of phyB”, instead of “.. activation of downstream phyB.”?

Response: This sentence has been changed to “PIF4, PIF5 and PIF7 have been implicated downstream of phyB”.

d. Line 234: “similar to their response to *bsk3* and *br1-301*”

Response: This sentence has been changed to “similar to the responses of *bsk3* and *bri1-301*”.

2. Line 111 refers to five RTFLs, without looking at the figure it is unclear about which five RTFLs you are mentioning.

Response: We have added the detailed RTFLs (*RTFL13*, *RTFL16*, *RTFL17*, *RTFL18*, and *RTFL21*).

3. You present data from previous work in figures 1a, supplemental figure 1b, and some other supplemental figures without a need for it. I suggest to restrict to quantitative RT-PCR data that you focus on mostly anyway.

Response: Supplemental Fig.1b was removed.

4. In figure 1c,d, and e, the Y-axis mentions a ratio, however, both components

of this ratio are plotted in the figure. This is impossible, the ratio per definition excludes the two bars.

Response: We have modified the legends of the Y-axis in Fig.1c (the expression of *RTFLs* under WL was standardized as “1”), d (the expression of *RTFLs* under dark conditions was standardized as “1”) and e (the expression of *RTFLs* in Col-0 was standardized as “1”) as relative expression. To show the expression differences of *RTFLs* under different conditions, we showed the groups with expression values normalized to 1. And significances between groups have been shown (Student's *t* test, * $P < 0.05$, ** $P < 0.01$, *** $P < 0.001$, ns indicates no significance).

5. In various figures you have provided phenotypic quantifications alongside with photographic images. This is a good idea, but it has become confusing: you are probably using the following order of genotypes as in the graph underneath, but in fact a direct comparison is impossible: in the graph you have WL and +FR directly coupled per genotype, whereas in the photos you first show all the genotypes in WL and then show all the genotypes in +FR. This is very confusing in Figure 3. I suggest you do it as you did in Fig.5a,b,d,e, but then include genotype names underneath your photos so as to be make it entirely clear what is what.

Response: As the reviewer suggested, we have changed the phenotypic pictures and added the genotype names underneath our photographic images in Fig. 3, Fig. 5, and Fig. 6.

6. It would help the reader to understand the line of reasoning if some choices are better supported by experimental data. For instance the selection of BKS3 is not explained, as it is missing from supplemental figure 4b.

Response: We found multiple BSKs from our IP-MS dataset. The *bsk3* single mutant has been reported to be sensitive to eBL treatment compared to other single mutants¹⁵, and BSK3 has been reported to be located on the cell membrane¹⁶. Therefore, we first tested the interaction between BSK3 and RTFL18.

7. In supplementary figure 5g, it is not clear if the bar graphs are quantifications of the western blots on the right, or if they are independent qRT-PCRs.

Response: In Supplementary Fig. 5g and h, the left bar charts represent the relative transcriptional levels of *BSK3/6* in Col-0, *bsk3/6*, *BSK3ox/BSK6ox*, and *dvl1-1D*BSK3ox/dvl1-1D*BSK6ox* by qRT-PCR. The right images represent the protein levels of *BSK3ox/BSK6ox* and *dvl1-1D*BSK3ox/dvl1-1D*BSK6ox* by western blotting. This information has been listed in the figure legend.

Reference

- 1 Wen, J., Lease, K. A. & Walker, J. C. DVL, a novel class of small polypeptides: overexpression alters Arabidopsis development. *The Plant journal : for cell and molecular biology* **37**, 668-677, doi:10.1111/j.1365-313x.2003.01994.x (2004).
- 2 Wen, J. & Walker, J. J. H. o. B. A. P. DVL Peptides are Involved in Plant Development. (2006).
- 3 Carabelli, M. *et al.* Canopy shade causes a rapid and transient arrest in leaf development through auxin-induced cytokinin oxidase activity. *Genes & development* **21**, 1863-1868, doi:10.1101/gad.432607 (2007).
- 4 Hayes, S. *et al.* Soil Salinity Limits Plant Shade Avoidance. *Current biology : CB* **29**, 1669-1676.e1664, doi:10.1016/j.cub.2019.03.042 (2019).
- 5 Wu, Z. *et al.* Functional and Structural Characterization of a Receptor-Like Kinase Involved in Germination and Cell Expansion in Arabidopsis. *Frontiers in plant science* **8**, 1999, doi:10.3389/fpls.2017.01999 (2017).
- 6 Bellstaedt, J. *et al.* A Mobile Auxin Signal Connects Temperature Sensing in Cotyledons with Growth Responses in Hypocotyls. *Plant physiology* **180**, 757-766, doi:10.1104/pp.18.01377 (2019).
- 7 Hamasaki, H. *et al.* Light Activates Brassinosteroid Biosynthesis to Promote Hook Opening and Petiole Development in Arabidopsis thaliana. *Plant & cell physiology* **61**, 1239-1251, doi:10.1093/pcp/pcaa053 (2020).
- 8 Wu, G. & Spalding, E. P. Separate functions for nuclear and cytoplasmic cryptochrome 1 during photomorphogenesis of Arabidopsis seedlings. *Proceedings of the National Academy of Sciences of the United States of America* **104**, 18813-18818, doi:10.1073/pnas.0705082104 (2007).
- 9 Minami, A., Takahashi, K., Inoue, S. I., Tada, Y. & Kinoshita, T. Brassinosteroid Induces Phosphorylation of the Plasma Membrane H⁺-ATPase during Hypocotyl Elongation in Arabidopsis thaliana. *Plant & cell physiology* **60**, 935-944, doi:10.1093/pcp/pcz005 (2019).
- 10 Narita, N. N. *et al.* Overexpression of a novel small peptide ROTUNDIFOLIA4 decreases cell proliferation and alters leaf shape in Arabidopsis thaliana. *The Plant journal : for cell and molecular biology* **38**, 699-713, doi:10.1111/j.1365-313X.2004.02078.x (2004).
- 11 Ikeuchi, M. *et al.* ROTUNDIFOLIA4 regulates cell proliferation along the body axis in Arabidopsis shoot. *Plant & cell physiology* **52**, 59-69, doi:10.1093/pcp/pcq138 (2011).
- 12 Li, L. *et al.* Linking photoreceptor excitation to changes in plant architecture. *Genes &*

- development* **26**, 785-790, doi:10.1101/gad.187849.112 (2012).
- 13 Yang, C. *et al.* Phytochrome A Negatively Regulates the Shade Avoidance Response by Increasing Auxin/Indole Acetic Acid Protein Stability. *Developmental cell* **44**, 29-41.e24, doi:10.1016/j.devcel.2017.11.017 (2018).
- 14 Huang, X. *et al.* Shade-induced nuclear localization of PIF7 is regulated by phosphorylation and 14-3-3 proteins in Arabidopsis. *eLife* **7**, doi:10.7554/eLife.31636 (2018).
- 15 Sreeramulu, S. *et al.* BSKs are partially redundant positive regulators of brassinosteroid signaling in Arabidopsis. *The Plant journal : for cell and molecular biology* **74**, 905-919, doi:10.1111/tpj.12175 (2013).
- 16 Ren, H. *et al.* BRASSINOSTEROID-SIGNALING KINASE 3, a plasma membrane-associated scaffold protein involved in early brassinosteroid signaling. *PLoS genetics* **15**, e1007904, doi:10.1371/journal.pgen.1007904 (2019).

REVIEWER COMMENTS

Reviewer #1 (Remarks to the Author):

In this revised edition, the authors have mostly adequately responded to my and other reviewer's comments. As a result, this manuscript is an improvement over its predecessor. Kudos to the authors for their quick addition of data and improvement of the writing. But there are still some uncertainties, and one of the new data turned out to be quite inadequate. Below are my concerns.

First, Figure R1 shown in reply to my rebuttal is completely false. See Figure R1b. Some show longitudinal sections of SAMs, others show longitudinal sections of leaf primordia. This clearly shows the author's lack of anatomical sense. Protrusions with proliferating or differentiated cells at their tips are not SAMs, but leaf primordia. The graph shown here has no scientific value, as it is judged to be a mixture of correct SAM cross-sections and partial leaf primordium cross-sections.

Second, 'genetic interactions' in page 10 is not sufficiently examined/proven. The authors wrote that "BSK3/6 functions downstream of RTFL18 and that RTFL18 negatively regulates the function of BSKs". But to say so, the phenotype of *dvl1-1D* must be compared with *BSKo/x* and *BSKo/x dvl1-1D*. But here the author compared *dvl1-1D* with only *BSKo / x dvl1-1D*. A genetic relationship between *DVL1* and *BSK* cannot be determined from this partial comparison. Moreover, and importantly, the *BSK3o/x* strains examined here are distinct from those used in combination with *dvl1-1D* (see Figures 5a-c and d-f). Because the *BSK3o/x* lines are not identical we cannot compare them.

Minor comments

- 1)) Line 80 and elsewhere, the term "silique": Here the authors are asked to use the more appropriate term "fruit". This is because the term "silique" is the name of the type of fruit, not the plant organ.
- 2) Line 95, "leaf and petiole": "leaf" includes leaf blade (leaf lamina), and petiole, so it would be better to write "leaf blade and petiole" here.
- 3) Fig. 1e. The authors used the "classic" *phyB-9* as material. However, as previously reported, *phyB-9* widely distributed in the past contains a second mutation that makes the 'phyB-like' phenotypes more severe (YOSHIDA et al. 2018). The purified and authentic *phyB-9* is now being distributed by the ABRC (stock number CS71624). The authors re-examine the *phyB*-related data using this.

Reviewer #2 (Remarks to the Author):

I would like to thank the authors for their consideration of the reviewers comments. I think this has resulted in a much more robust manuscript and I believe it will be warmly welcomed by the field.

Most of my previous comments have been addressed, but I still have a query about the statistics used for the qPCRs. Are the T-tests used in figure 1 corrected for multiple comparisons?

In addition to this I noticed that in the revised methods there are a couple of instances of "X technique was performed as in study Y". To ensure the reproducibility of these results it is essential that the authors specify within this manuscript how these experiments were performed.

Once these issues have been addressed I would happily recommend this manuscript for publication.

Reviewer #3 (Remarks to the Author):

Huang et al have made substantial, helpful revisions of their manuscript and study an interesting question of how RTFL peptides contribute to shade avoidance in *Arabidopsis*. The data provides new insights into shade avoidance co-regulation by these peptides and the authors use an elegant combination of approaches.

The authors have included substantial data that required to strengthen their conclusions and

interpretations. I do have some remaining points that were not resolved in the revisions.

Previous points:

RTFL18-BSK3/6-PIF4 pathway in Abstract and throughout: this problem was not resolved. Why insist on keeping this unchanged, while providing data that show no involvement of RTFL18 other than as one of many genes? There is no good reason for this focus on RTFL18, and you convincingly show they should be considered together, which also means that some of the studies should be done on more than just one member. RTFL18 gene expression is most strongly induced by FR, as compared to the other RTFLs, and this is your reason to do all work on this particular family member, yet its knockout has a completely wildtype shade avoidance phenotype. I am therefore not convinced that this validates putting RTFL18 central.

The RTFL18 functional involvement is thus based entirely on overexpression data, and on the combined higher order mutant phenotype. The description of the module should thus be extended to other RTFLs, for example RTFL18/21-BSK3/6-PIF4 pathway, but that would require obtaining data for other RTFLs, as was done for RTFL18. The easiest solution is probably to change the naming of this module into one that does not specify a particular RTFL family member for as long as you have no definitive proof for this member to have a functional role.

Tissue-specificity of qRT-PCR and ChIP-qPCR: rather than including data that are tissue-specific, consistent with your other data, you did the opposite and included some bulk data in Figures 6 and 7. This is not exactly resolving this point. It is good to see though in revised Fig. 2b that RTFL18 is FR-inducible in all shoot tissues. My point was that regardless of ubiquitous RTFL18 gene expression upregulation by FR in all shoot tissues, functional interactions may not be similar between all tissues. If one tissue expresses less of a putative interactor than another, the interactions in bulk tissues may not represent in vivo interactions in the organ that you make inferences about. Hence the suggestion to work more tissue specifically, rather than less specific.

RTFL18 localisation: You added nice data confirming the PM localisation indeed. I do have a question about the Fig. 2e upper panel (anti-FLAG): there are two non-PM hotspots of signal in the left upper and right bottom part of the image. Do you have any idea what this is? It doesn't seem to be inside the cells in focus, but the image also does not seem to be a projection of multiple layers.

Authors' Responses to Reviewers' Comments (NCOMMS-23-03409A)

Reviewer #1 (Remarks to the Author):

In this revised edition, the authors have mostly adequately responded to my and other reviewer's comments. As a result, this manuscript is an improvement over its predecessor. Kudos to the authors for their quick addition of data and improvement of the writing. But there are still some uncertainties, and one of the new data turned out to be quite inadequate. Below are my concerns.

Response: We thank Reviewer #1 for the positive consideration of this work.

First, Figure R1 shown in reply to my rebuttal is completely false. See Figure R1b. Some show longitudinal sections of SAMs, others show longitudinal sections of leaf primordia. This clearly shows the author's lack of anatomical sense. Protrusions with proliferating or differentiated cells at their tips are not SAMs, but leaf primordia. The graph shown here has no scientific value, as it is judged to be a mixture of correct SAM cross-sections and partial leaf primordium cross-sections.

Response: We apologized for our mistakes on the pervious pictures of SAM. We have consulted with an expert of SAM (Prof. Weibing Yang in University of Chinese Academy of Sciences) and selected the right images of SAM in Fig. R1. Compared to Col-0 and *rtfl18*rtfl21*, *dvl1-1D* has smaller SAM.

Fig. R1 | SAM of Col-0, *dvl1-1D* and *rtfl18*rtfl21* under WL and low R:FR conditions. The left panels show SAM images. The scale bar represents 50 µm.

The bar graph represents the height of SAM. Seedlings grown for 4 days under white light were transferred to low R:FR conditions or remained under white light for 1 day. The error bars indicate the SEM. The letters indicate significant differences between mean values (two-way ANOVA: Tukey's multiple comparisons test, $P < 0.05$), and groups with the same letters are not significantly different.

Second, 'genetic interactions' in page 10 is not sufficiently examined/proven. The authors wrote that "BSK3/6 functions downstream of RTFL18 and that RTFL18 negatively regulates the function of BSKs". But to say so, the phenotype of *dvl1-1D* must be compared with BSKo/x and BSKo/x *dvl1-1D*. But here the author compared *dvl1-1D* with only BSKo / x *dvl1-1D*. A genetic relationship between DVL1 and BSK cannot be determined from this partial comparison. Moreover, and importantly, the BSK3o/x strains examined here are distinct from those used in combination with *dvl1-1D* (see Figures 5a–c and d–f). Because the BSK3o/x lines are not identical we cannot compare them.

Response: We agree with the reviewer that we would better to compare the phenotype between *dvl1-1D*, *BSK3ox*, *BSK6ox*, *dvl1-1D * BSK3ox*, and *dvl1-1D * BSK6ox*. However, we didn't screen single insertion homozygous lines for *BSK3/6ox*. Although we got *BSK3/6ox* hybridized with *dvl1-1D* by crossing *BSK3/6ox* with *dvl1-1D*, and we performed western blotting to confirm the expression of BSKs and genotyping for *dvl1-1D*, the *BSK3/6ox* lines are not identical in *dvl1-1D * BSK3/6ox* due to the genetic segregation. We tried to do the comparisons in Fig. R2 and can see that the petiole lengths of the cotyledon and the first and second leaves in *BSK3/6ox* and *dvl1-1D * BSK3/6ox* were significantly longer than that in *dvl1-1D*, which was consistent with the results presented in Fig. 5a-c, d-f. The petiole lengths of *BSK3/6ox* were longer than that in *dvl1-1D * BSK3/6ox* because of the negative effect of overexpressed RTFL18 on the function of BSKs. This result supported our conclusion, but it is not rigorous. So we didn't put this result in our manuscript and we modified the

related sentence as “BSK3/6 may function downstream of RTFL18.” (Line 251)

Fig. R2 | The phenotype of *dvl1-1D*, *BSKox*, and *dvl1-1D*BSKox*. a, The lengths of the cotyledon petiole, the first and second petiole, and hypocotyl in Col-0, *dvl1-1D*, *BSK3ox* (#8, #16), and *dvl1-1D*BSK3ox* (#8, #16) lines. b, The lengths of the cotyledon petiole, the first and second petiole, and hypocotyl in Col-0, *dvl1-1D*, *BSK6ox* (#26, #25), and *dvl1-1D*BSK3ox* (#26, #25) lines. Seedlings grown for 3 days under white light were transferred to low R:FR conditions or remained under white light for 4/8 days. The error bars indicate the SEM (n>13). Letters indicate significant differences between mean values (two-way ANOVA: Tukey’s multiple comparisons test, $P < 0.01$), and groups with the same letters are not significantly different. Scale bars represent 5 mm.

Minor comments

1)) Line 80 and elsewhere, the term "silique": Here the authors are asked to use the more appropriate term "fruit". This is because the term "silique" is the

name of the type of fruit, not the plant organ.

Response: Corrected (Line 80, 132).

2) Line 95, "leaf and petiole": "leaf" includes leaf blade (leaf lamina), and petiole, so it would be better to write "leaf blade and petiole" here.

Response: Corrected (Line 95).

3) Fig. 1e. The authors used the "classic" *phyB-9* as material. However, as previously reported, *phyB-9* widely distributed in the past contains a second mutation that makes the 'phyB-like' phenotypes more severe (YOSHIDA et al. 2018). The purified and authentic *phyB-9* is now being distributed by the ABRC (stock number CS71624). The authors re-examine the *phyB*-related data using this.

Response: We have got the *phyB-9^{BC}* and replaced RT-qPCR data in Fig. 1e.

Fig. R3 | Peak plot of sequencing results for *PHYB* and *VEN4* in Col-0, *phyB-9^{BC}* (with mutation in *PHYB*, but without the mutation in *VEN4*) and *phyB-9*.

Reviewer #2 (Remarks to the Author):

I would like to thank the authors for their consideration of the reviewers comments. I think this has resulted in a much more robust manuscript and I believe it will be warmly welcomed by the field.

Response: We appreciated the reviewer #2' comments.

Most of my previous comments have been addressed, but I still have a query

about the statistics used for the qPCRs. Are the T-tests used in figure 1 corrected for multiple comparisons?

Response: Thanks for pointing this out. We conducted *t*-test with multiple comparison corrections in revised Fig. 1c, d, e, and h. The related information has been added in figure legend and methods (Line 622-626).

In addition to this I noticed that in the revised methods there are a couple of instances of "X technique was performed as in study Y". To ensure the reproducibility of these results it is essential that the authors specify within this manuscript how these experiments were performed.

Response: Thanks, we have added more detailed information in Materials and methods, including ChIP assay (Line 468-485), Bimolecular fluorescent complementation (BiFC) assay (Line 515-516), RNA-seq library preparation, construction, and analysis (Line 593-602), and Western blotting and analysis (Line 638-643).

Once these issues have been addressed I would happily recommend this manuscript for publication.

Reviewer #3 (Remarks to the Author):

Huang et al have made substantial, helpful revisions of their manuscript and study an interesting question of how RTFL peptides contribute to shade avoidance in Arabidopsis. The data provides new insights into shade avoidance co-regulation by these peptides and the authors use an elegant combination of approaches.

Response: We thank Reviewer #3 for the positive consideration.

The authors have included substantial data that required to strengthen their conclusions and interpretations. I do have some remaining points that were not resolved in the revisions.

Previous points:

RTFL18-BSK3/6-PIF4 pathway in Abstract and throughout: this problem was

not resolved. Why insist on keeping this unchanged, while providing data that show no involvement of RTFL18 other than as one of many genes? There is no good reason for this focus on RTFL18, and you convincingly show they should be considered together, which also means that some of the studies should be done on more than just one member. RTFL18 gene expression is most strongly induced by FR, as compared to the other RTFLs, and this is your reason to do all work on this particular family member, yet its knockout has a completely wildtype shade avoidance phenotype. I am therefore not convinced that this validates putting RTFL18 central.

The RTFL18 functional involvement is thus based entirely on overexpression data, and on the combined higher order mutant phenotype. The description of the module should thus be extended to other RTFLs, for example RTFL18/21-BSK3/6-PIF4 pathway, but that would require obtaining data for other RTFLs, as was done for RTFL18. The easiest solution is probably to change the naming of this module into one that does not specify a particular RTFL family member for as long as you have no definitive proof for this member to have a functional role.

Response: Thanks for the suggestion. We have used RTFLs instead of RTFL18 in main text (line 28, 164, 182, 189, 327, and 394). To further support the function redundancy of RTFLs, we observed the phenotypes (cotyledon petiole length, leaf petiole length and hypocotyl length) of *35S::RTFL21* and *35S::RTFL21-Flag* transgenic lines. The *RTFL21* overexpression transgenic lines displayed similar phenotypes to *RTFL18* overexpression plants (revised Supplementary Fig. 3b, e). We also used LCI assay to determine the interaction between RTFL21 and BSK3/6. RTFL21 can interact with BSK3/6, similar to RTFL18 results (revised Supplementary Fig. 4c).

Tissue-specificity of qRT-PCR and CHIP-qPCR: rather than including data that are tissue-specific, consistent with your other data, you did the opposite and included some bulk data in Figures 6 and 7. This is not exactly resolving this

point. It is good to see though in revised Fig. 2b that RTFL18 is FR-inducible in all shoot tissues. My point was that regardless of ubiquitous RTFL18 gene expression upregulation by FR in all shoot tissues, functional interactions may not be similar between all tissues. If one tissue expresses less of a putative interactor than another, the interactions in bulk tissues may not represent *in vivo* interactions in the organ that you make inferences about. Hence the suggestion to work more tissue specifically, rather than less specific.

Response: We agree with the reviewer about the tissue specific studies. We added low R:FR inductions of more *RTFLs* in different tissues (revised Supplementary Fig. 2b). Low R:FR induced the expressions of *RTFL13/16/21* in hypocotyls, cotyledons & petioles, and leaf & petioles, while *RTFL17* was mainly induced in cotyledons & petioles (revised Supplementary Fig. 2b). These induction patterns suggest that different RTFL/DVL peptides may function in different tissues under low R: FR.

We also agree that functional interactions of RTFLs with putative interactors, such as BSKs, may not be similar between all tissues. The tissue expression patterns of BSK3 and BSK6 were shown in Supplementary Fig. 5a and 5b. The tissue specific expression levels of these putative interactors will also infer *in vivo* interactions. More tissue specific studies are required in future. We added these sentences in discussion (line 339-343).

RTFL18 localisation: You added nice data confirming the PM localisation indeed. I do have a question about the Fig. 2e upper panel (anti-FLAG): there are two non-PM hotspots of signal in the left upper and right bottom part of the image. Do you have any idea what this is? It doesn't seem to be inside the cells in focus, but the image also does not seem to be a projection of multiple layers.

Response: We considered non-PM hotspots of signal may be unlabeled free fluorescein. To support our conclusion, we presented more immunofluorescence images in revised Supplementary Fig. 2c.

REVIEWERS' COMMENTS

Reviewer #1 (Remarks to the Author):

In this revision, the authors improved the text and some Supplemental data. Now I found only a few points to be fixed as listed below.

1) The relationship between *dvl1-1D*, *BSKox* and *dvl1-1D BSKox*. Here the authors showed the comparison among them in Fig. R2 in the response letter. From these data, it can be said: *dvl1-1D* has shorter leaves both under white light and SAS-inducible conditions; *BSKox* showed enhanced elongation only under SAS condition; and *dvl1-1D BSKox* showed intermediate phenotype between *dvl1-1D* and *BSKox* under SAS. Genetically this data is to be interpreted that the relationship between DVL1 pathway and BSK pathway is 'independent'. This would be reasonable because in the authors' scheme DVL1 (RTFLs) and BSKs proteins interact to function to regulate SAS. In this case, the BSKs are NOT 'downstream' of DVL1 (RTFLs). The term 'downstream' must be used for the target of the BSKs-RTFL complex, namely, in this scheme, PIFs. Therefore, the description in Line 252 should be revised to consider the above point. The discussion part corresponding to this point is also to be revised.

2) As the other reviewer pointed, the term 'leaf' should be correctively changed to 'leaf blade(s)' also in Lines 134 and 137.

3) Discussion part. The meaning of 'suppression' of SAS by RTFLs should be more carefully discussed. In Lines 320, 354-358, and 395. To make the suppression to be meaningful, the RTFL-BSK module must work significantly LATER than activation of SAS. Some time lag must be expected for it. From molecular mechanism that the authors think it could be rationalized.

4) Reference. As pointed before, reference list should be more carefully checked. Some titles are in a different format from others (for example, every word started with large capitals); Latin name must be in italic for reference #26; there is no title, year, journal name or so on in reference number 28; Journal name is too precise for *The Plant Journal* (subtitle of this journal is not needed) and so on.

Authors' Responses to Reviewers' Comments

Reviewer #1 (Remarks to the Author):

In this revision, the authors improved the text and some Supplemental data.

Now I found only a few points to be fixed as listed below.

1) The relationship between *dvl1-1D*, *BSKox* and *dvl1-1D BSKox*. Here the authors showed the comparison among them in Fig. R2 in the response letter. From these data, it can be said: *dvl1-1D* has shorter leaves both under white light and SAS-inducible conditions; *BSKox* showed enhanced elongation only under SAS condition; and *dvl1-1D BSKox* showed intermediate phenotype between *dvl1-1D* and *BSKox* under SAS. Genetically this data is to be interpreted that the relationship between DVL1 pathway and BSK pathway is 'independent'. This would be reasonable because in the authors' scheme DVL1 (RTFLs) and BSKs proteins interact to function to regulate SAS. In this case, the BSKs are NOT 'downstream' of DVL1 (RTFLs). The term 'downstream' must be used for the target of the BSKs-RTFL complex, namely, in this scheme, PIFs. Therefore, the description in Line 252 should be revised to consider the above point. The discussion part corresponding to this point is also to be revised.

Response: We have removed "downstream" descriptions and revised the relevant sentences in manuscript.

2) As the other reviewer pointed, the term 'leaf' should be correctively changed to 'leaf blade(s)' also in Lines 134 and 137.

Response: Corrected.

3) Discussion part. The meaning of 'suppression' of SAS by RTFLs should be more carefully discussed. In Lines 320, 354-358, and 395. To make the suppression to be meaningful, the RTFL-BSK module must work significantly LATER than activation of SAS. Some time lag must be expected for it. From molecular mechanism that the authors think it could be rationalized.

Response: We have added the relation discussion in line 356-362.

4) Reference. As pointed before, reference list should be more carefully

checked. Some titles are in a different format from others (for example, every words started with large capitals); Latin name must be in italic for reference #26; there is no title, year, journal name or so on in reference number 28; Journal name is too precise for The Plant Journal (subtitle of this journal is not needed) and so on.

Response: Thanks. We have rechecked and revised the issues in the references.